



# Efficient multi-angle polarimetric inversion of aerosols and ocean color powered by a deep neural network forward model

Meng Gao[1,2], Bryan A. Franz[1], Kirk Knobelspiesse[1], Peng-Wang Zhai[3], Vanderlei Martins[3], Sharon Burton[4], Brian Cairns[5], Richard Ferrare[4], Joel Gales[1,6], Otto Hasekamp[7], Yongxiang Hu[4], Amir Ibrahim[1,2], Brent McBride[3], Anin Puthukkudy[3], P. Jeremy Werdell[1], and Xiaoguang Xu[3]

[1]NASA Goddard Space Flight Center, Code 616, Greenbelt, Maryland 20771, USA
[2]Science Systems and Applications, Inc., Greenbelt, MD, USA
[3]JCET/Physics Department, University of Maryland, Baltimore County, Baltimore, MD 21250, USA
[4]MS 475 NASA Langley Research Center, Hampton, VA 23681-2199, USA
[5]NASA Goddard Institute for Space Studies, New York, NY 10025, USA
[6]Science Applications International Corp., Greenbelt, MD, USA
[7]Netherlands Institute for Space Research (SRON, NWO-I), Utrecht, The Netherlands

**Correspondence:** Meng Gao (meng.gao@nasa.gov)

**Abstract.** NASA's Plankton, Aerosol, Cloud, ocean Ecosystem (PACE) mission, scheduled for launch in the timeframe of 2023, will carry a hyperspectral scanning radiometer named the Ocean Color Instrument (OCI) and two Multi-Angle Polarimeters (MAP): the UMBC Hyper-Angular Rainbow Polarimeter (HARP2) and the SRON Spectro-Polarimeter for Planetary EXploration one (SPEXone). The MAP measurements contain rich information on the microphysical properties of aerosols and

hydrosols, and therefore can be used to retrieve accurate aerosol properties for complex atmosphere and ocean systems. Most polarimetric aerosol retrieval algorithms utilize vector radiative transfer models iteratively in an optimization approach, which leads to high computational costs that limit their usage in the operational processing of large data volumes acquired by the MAP imagers. In this work, we propose a deep neural network (NN) model to represent the radiative transfer simulation of coupled atmosphere and ocean systems, for applications to the HARP2 instrument and its predecessors. Through the evaluation

of synthetic datasets for AirHARP (airborne version of HARP2), the NN model achieves a numerical accuracy smaller than the instrument uncertainties, with a running time of 0.01s in a single CPU core or 1 ms in GPU. Using the NN as a forward model, we built an efficient joint aerosol and ocean color retrieval algorithm called FastMAPOL, evolved from the well-validated Multi-Angular Polarimetric Ocean coLor (MAPOL) algorithm. Retrievals of aerosol properties and water leaving signals were conducted on both the synthetic data and the AirHARP field measurements from the Aerosol Characterization from Polarimeter and Lidar (ACEPOL) campaign in 2017. From the validation with the synthetic data and the collocated High Spectral Resolu-

tion Lidar (HSRL) aerosol products, we demonstrated that the aerosol microphysical properties and water leaving signals can be retrieved efficiently and within acceptable error. The FastMAPOL algorithm can be used to operationally process the large volume of polarimetric data acquired by PACE and other future Earth observing satellite missions with similar capabilities.



# 1 Introduction

Atmospheric aerosols are tiny particles suspended in the atmosphere, such as dust, sea salt, and volcanic ash, that play important roles in air quality (Shiraiwa et al., 2017; Li et al., 2017) and Earth's climate (Boucher et al., 2013). Aerosols influence the Earth's reflectivity directly through scattering and absorption of solar light, and indirectly through interactions with clouds. The radiative forcing from aerosols is one of the main uncertainties in studies of global climate change (Boucher et al., 2013). When deposited into ocean waters, aerosols also contribute to the availability of nutrients needed for phytoplankton growth, and thereby influence ocean ecosystems (Westberry et al., 2019). Accurate knowledge of aerosol optical properties is also important for atmospheric correction in ocean color remote sensing, wherein the spectral water leaving radiances are retrieved by subtracting the contributions of the atmosphere and ocean surface from the spaceborne or airborne measurements made at the top of atmosphere (Mobley et al., 2016). The resulting water leaving signals provide valuable information to derive biogeochemical quantities for monitoring the global ocean ecosystem (Dierssen and Randolph, 2013), and quantifying ocean biochemical processes (Platt et al., 2008). Accurate assessments of aerosol optical and microphysical properties are thus important for both atmospheric and oceanic studies.

Multi-angle polarimeters (MAPs) measure polarized light at continuous or discrete spectral bands and at multiple viewing angles, providing rich information on aerosol optical and microphysical properties (Mishchenko and Travis, 1997; Chowdhary et al., 2001; Hasekamp and Landgraf, 2007; Knobelspiesse et al., 2012). The Polarization and Directionality of the Earth's Reflectance (POLDER) instrument pioneered the spaceborne MAP, which was hosted on Advanced Earth Observing Satellite missions (ADEOS-I; 1996-1997 and ADEOS-II; 2002-2003), and Polarization and Anisotropy of Reflectances for Atmospheric Sciences Coupled with Observations from a Lidar (PARASOL; 2004-2013) mission (Tanré et al., 2011). The Hyper-Angular Rainbow Polarimeter (HARP) CubeSat, a small satellite with 3U volume, was launched from the International Space Station on Feburary of 2020 and has captured scientific images (UMBC Earth and Space Institute). There are several satellite missions with MAP instruments scheduled to be launched in the time frame of 2023-2024, including ESA's Multi-viewing Multi-channel Multi-polarisation Imager (3MI) (Fougnie et al., 2018) and NASA's Multi-Angle Imager for Aerosols (MAIA) (Diner et al., 2018) and Plankton, Aerosol, Cloud, ocean Ecosystem (PACE) (Werdell et al., 2019) missions. A thorough review of the MAP instruments and algorithms can be found in Dubovik et al. (2019).

The PACE mission will carry a hyperspectral scanning radiometer named the Ocean Color Instrument (OCI) and two MAPs: a next generation UMBC (University of Maryland, Baltimore County) Hyper-Angular Rainbow Polarimeter (HARP2) (Martins et al., 2018), and the SRON (Netherlands Institute for Space Research) Spectro-Polarimeter for Planetary EXploration one (SPEXone) (Hasekamp et al., 2019a). OCI will provide continuous spectral measurements from the ultraviolet (340 nm) to near-infrared (890 nm) with Full Width Half Maximum of 5 nm resolution and sampling every 2.5 nm, plus a set of seven discrete shortwave infrared (SWIR) bands centered at 940, 1038, 1250, 1378, 1615, 2130, and 2260 nm. SPEXone performs multiangle measurements at 5 along track viewing angles of $0°$, $\pm20°$ and $\pm58°$, with a surface swath of 100km, and a continuous spectral range spanning 385-770 nm at resolutions of 2-3 nm for intensity and 10-40 nm for polarization (Rietjens et al., 2019). HARP2 is a wide field-of-view imager that measures the polarized radiances at 440, 550, 670, and 870 nm,





where the 670 nm band will measure 60 viewing angles and the other bands 10 viewing angles, with a swath of 1,556 km at nadir on the Earth surface. To facilitate cross calibrations and validations, a PACE Level-1C common data format has been developed, with the purpose of projecting all three PACE instruments onto an uniform spatial grid (Plankton, Aerosol, Cloud, ocean Ecosystem (PACE) mission, 2020). The PACE instruments will provide an unprecedented opportunity to improve the

characterization of the atmosphere and ocean states (Remer et al., 2019a, b; Frouin et al., 2019).

To retrieve the aerosol information from polarimetric measurements over oceans, several advanced aerosol retrieval algorithms have been developed for both airborne and spaceborne MAPs, such as POLDER/PARASOL(Hasekamp et al., 2011; Dubovik et al., 2011, 2014; Li et al., 2019; Hasekamp et al., 2019b; Chen et al., 2020), the Airborne Multiangle SpectroPolarimetric Imager (AirMSPI) (Xu et al., 2016, 2019), SPEX Airborne (the airborne version of SPEXone ) (Fu and Hasekamp,

2018; Fu et al., 2020; Fan et al., 2019), the Research Scanning Polarimeter (RSP) (Chowdhary et al., 2005; Wu et al., 2015; Stamnes et al., 2018; Gao et al., 2018, 2019, 2020), the Directional Polarimetric Camera (DPC)/GaoFen-5 (Wang et al., 2014; Li et al., 2018). The retrieval algorithms are mostly based on iterative optimization approaches that utilize vector radiative transfer (RT) models as the forward model. The high computational cost of the RT simulations pose great challenges in the operational processing of the large data volumes acquired by the MAP imagers. To alleviate this issue, the SPEX team repre-

sented the polarimetric reflectance for an open ocean system using a deep neural network (NN) and coupled it with a radiative transfer model for the atmosphere (Fan et al., 2019). This hybrid forward model avoids the direct calculation of the scattering and absorption properties inside the ocean, and still maintains high accuracy, therefore enabling sufficient efficiency for SPEXone data retrieval. For coastal waters, Mukherjee et al. (2020) developed a NN model to predict the polarimetric reflectance associated with complex water optical properties. This NN model can be combined with a flexible atmosphere model for MAP

aerosol retrievals over complex waters.

For non-polarimetric remote sensing studies, several NN approaches have been developed to derive aerosol and ocean properties simultaneously (Fan et al. (2017); Shi et al. (2020) and reference within). Fan et al. (2017) developed NN models to directly invert the aerosol optical depth (AOD) and remote sensing reflectance $R_{rs}(\lambda)$ $(sr^{-1})$ from the NASA Moderate Resolution Imaging Spectroradiometer (MODIS) measurements. Shi et al. (2020) developed a NN radiative transfer scheme for

coupled atmosphere and ocean systems including both open and coastal waters, which is then applied in an optimal estimation algorithm for the Cloud and Aerosol Imager-2 (CAI-2) hosted on the Greenhouse gases Observing Satellite-2 (GOSAT-2).

A number of NN models have been developed to directly invert the aerosol microphysical properties from MAP measurements. Di Noia et al. (2015) discusses the NN employed to retrieve aerosol refractive index, size, and optical depth (AOD) from GroundSPEX (a ground version of SPEX instrument) measurements. Di Noia et al. (2017) developed a NN inversion method

for airborne MAP measurement over land from RSP. In both works, the results from the NN inversion are further used as initial values for iterative optimization, and both efficiency and the retrieval accuracy are shown to be improved. Using NN to conduct direct inversion is efficient, but it is often viewed as a black box and it is difficult to account for measurement uncertainties. The combination of a NN inversion with an iterative optimization method shows promise for MAP retrievals.

Even with such ample progress, it is still challenging for current state-of-the-art algorithms to process MAP data opera-

tionally through iterative optimization. In this work, we present a joint retrieval algorithm for aerosol properties and water



leaving signals that uses a deep NN model to replace the radiative transfer forward model for simulation of the polarimetric reflectances. This approach is one step further than (Fan et al., 2019), as both the atmospheric and oceanic radiative transfer processes are represented by the NN. The NN forward model is then used in an iterative retrieval algorithm that is significantly more computationally efficient than approaches that use traditional radiative transfer. The benefits using a NN model as the

forward model in retrieval algorithms can be summarized as follows with details provided in later sections:

- Fast: NN models mostly involve matrix operations that can be evaluated efficiently.

- Accurate: Given sufficient training data volumes and accuracies, NN models can be trained with high precision.

- Differentiable: The Jacobian matrix of NN models can be represented analytically and therefore further improves efficiency and accuracy in retrievals.

- Transferable: The parameters of a NN can be exported and implemented into existing retrieval algorithms.

The retrieval algorithm we developed is called FastMAPOL, which is evolved from the well-validated Multi-Angular Polarimetric Ocean coLor (MAPOL) algorithm (Gao et al., 2018, 2019, 2020) by replacing its forward model with NN models. To validate the retrieval algorithm, we applied FastMAPOL to both synthetic and field measurements from AirHARP (the airborne version of HARP2 and HARP CubeSat) for the Aerosol Characterization from Polarimeter and Lidar (ACEPOL) campaign in

2017 (Knobelspiesse et al., 2020). The synthetic AirHARP data is a supplement of the field measurements with a wider range of aerosol and ocean optical properties, and solar and viewing geometries. The AOD derived from coincident High Spectral Resolution Lidar (HSRL, Hair et al. (2008)) and Aerosol Robotic Network (AERONET, Holben et al. (1998)) measurements are used to evaluate the performance of the AOD retrieval from the AirHARP field measurements. Using the retrieved aerosol properties, atmospheric correction is applied to the AirHARP measurements to derive the water leaving signal at four AirHARP

bands. The retrieved aerosol products from MAP can also assist hyperspectral atmospheric correction on instruments such as PACE OCI as previously demonstrated using the aerosol properties retrieved from RSP and hyperspectral measurements from SPEX Airborne (Gao et al., 2020; Hannadige et al., 2020). Retrieval uncertainties of both aerosol and water leaving signals under various aerosol loadings are also discussed in this study. The retrieval algorithm powered by the NN forward model provides a practical approach for operational applications of polarimetric aerosol and ocean color retrieval for PACE, and other

satellite missions that utilize polarimeters in the retrieval of geophysical properties from Earth observations.

The paper is organized into seven sections:, Sect. 2 reviews the retrieval algorithm and its radiative transfer forward model, Sect. 3 discusses the training and accuracy of the NN forward model, Sec 4. applies the NN forward model to aerosol and water leaving signal retrievals from the synthetic AirHARP data, Sect. 5. discusses the retrievals on AirHARP field measurements from ACEPOL campaign, Sect. 6 and 7 provide discussions and conclusions.

## 2   Joint aerosol and ocean color retrieval algorithm

In this section, we will discuss the MAPOL retrieval algorithm based on multi-angle polarimetric measurements and the associated radiative transfer forward model. The retrieval algorithm have been validated using both synthetic data (Gao et al.,





2018) and RSP field measurements (Gao et al., 2019, 2020). To apply the retrieval algorithm to AirHARP measurements, we will first discuss the AirHARP instrument characteristics.

AirHARP measures the total and linearly polarized radiance at 60 viewing angles at the 660 nm band, and at 20 viewing angles at the 440, 550, and 870 nm bands. Different from AirHARP, HARP2 reduces the number of viewing angles to 10 at

440, 550, and 870 nm, and maintains 20 viewing angles at 660 nm, in order to fulfill the bandwidth requirement and preserve information content as much as possible. HARP instruments (AirHARP, HARP CubeSat, and HARP2) use a modified three-way Phillips prism located after the front lens to split the incident light into the three orthogonal linear polarization states ($0°$, $45°$, and $90°$), which can be recombined to obtain the Stokes parameters $L_t$, $Q_t$, and $U_t$ at the aircraft altitude (Puthukkudy et al., 2020). Circular polarization (Stokes parameter V) is not measured by any of the polarimeters in ACEPOL as it is

negligible for atmospheric studies (Kawata, 1978). We use the total measured reflectance ($\rho_t(\lambda)$) and DoLP ($P_t(\lambda)$) at the height of the aircraft with spectral dependencies hereafter implied, which are defined as

$$\rho_t = \frac{\pi r^2 L_t}{\mu_0 F_0}, \tag{1}$$

$$P_t = \frac{\sqrt{Q_t^2 + U_t^2}}{L_t}, \tag{2}$$

where $F_0$ is the extraterrestrial solar irradiance, $\mu_0$ is the cosine of the solar zenith angle, $r$ is the Sun-Earth distance correction

factor in astronomical units.

Based on the MAP measurements, the MAPOL retrieval algorithm is developed to derive both the aerosol properties and the water leaving signal simultaneously. The retrieval algorithm minimizes the difference between the MAP measurements and the forward model simulations computed from vector radiative transfer simulations (Zhai et al., 2009, 2010). By assuming the measurement and modeling uncertainties follow Gaussian statistical distributions, the retrieval parameters can be estimated

through Bayesian theory using the cost function $\chi^2$ to quantify the difference between the measurement and the forward model simulation (Rogers, 2000):

$$\chi^2(\mathbf{x}) = \frac{1}{N} \sum_i \left( \frac{[\rho_t(i) - \rho_t^f(\mathbf{x};i)]^2}{\sigma_\rho^2(i)} + \frac{[P_t(i) - P_t^f(\mathbf{x};i)]^2}{\sigma_P^2(i)} \right) \tag{3}$$

where $\rho_t$ and $P_t$ are the measured reflectance and DoLP as defined in Eqs. (1) and (2), and $\rho_t^f$ and $P_t^f$ are the corresponding quantities computed from the forward model. The state vector $\mathbf{x}$ contains all retrieval parameters, such as the aerosol size,

refractive indices; the subscript $i$ stands for the index of the measurements at different viewing angles and wavelengths; and N is the total number of the measurements used in the retrieval. For AirHARP measurements, the maximum value of N is 120 considering all the viewing angles from the four bands. The total uncertainties of the reflectance and DoLP used in the algorithm are denoted as $\sigma_\rho$ and $\sigma_P$, which are contributed by both the measurement uncertainties $\sigma_m$ and the forward model uncertainties $\sigma_f$ (more details in Sect). 3.3:

$$\sigma_\rho^2 = \sigma_{\rho,m}^2 + \sigma_{\rho,f}^2 \tag{4}$$

$$\sigma_P^2 = \sigma_{P,m}^2 + \sigma_{P,f}^2 \tag{5}$$





One important component of $\sigma_m$ is the calibration uncertainty. AirHARP was calibrated in the lab with an accuracy of 3-5 % for reflectance, and 0.005 for DoLP (McBride et al., 2019). In-flight uncertainty for the AirHARP DoLP is conservatively estimated to be at most 0.01 without an onboard calibrator. In this study, we adopted the calibration uncertainty for reflectance as $\sigma_{\rho,cal} = 3\%\rho_t$ and for DoLP as $\sigma_{P,cal} = 0.01$ for all four bands. The accuracy of the HARP CubeSat and HARP2 mea-

surements can be further improved through onboard calibration (McBride et al., 2020; Puthukkudy et al., 2020). AirHARP conducted high spatial resolution measurements with a grid size of 55 m. We averaged every 10x10 pixels together (a box of 550m × 550 m). The standard deviations of the pixels within the box are used to estimate the random noise and the spatial variability of the geophysical properties ($\sigma_{avg}$). To account for the total measurement uncertainties, we considered the contributions from both the calibration ($\sigma_{cal}$) and pixel averaging ($\sigma_{avg}$) as

$$\sigma_m^2 = \sigma_{cal}^2 + \sigma_{avg}^2 \qquad (6)$$

for both reflectance and DoLP. As observed by AirHARP (Puthukkudy et al., 2020) and RSP measurements (Gao et al., 2020), the sunglint angular pattern cannot be well modeled by an isotropic Cox-Munk model. To minimize the impact of sunglint in our discussions, we removed the signals within an angle range of 0° to 40° relative to the solar specular reflection direction. Furthermore, noise correlation is an import influence on the retrieval accuracy (Knobelspiesse et al., 2012) that is ignored in

this study due to the lack of knowledge on this characteristic for AirHARP.

The forward model uncertainties $\sigma_f$ include the numerical accuracy of the radiative transfer calculation, and can include any estimation of uncertainties due to the incompleteness of the model to describe the system. For convenience, as discussed in the next section, we will only consider the uncertainty of the NN forward model ($\sigma_{NN}$) and the numerical accuracy of the radiative transfer simulation used for generating the NN training data ($\sigma_{RT}$):

$$\sigma_f^2 = \sigma_{RT}^2 + \sigma_{NN}^2. \qquad (7)$$

To fully utilize the information contained in the AirHARP measurements, the forward model needs to achieve an accuracy level much better than the measurement uncertainties. This becomes the goal of the NN training in the next section, to reproduced the forward model within an error that is much less than the measurement uncertainty. Detailed comparisons of the forward model uncertainties and the measurement uncertainties will be provided in the next section. To minimize the cost

function defined in Eq. (3), we use an optimization method, called the Subspace Trust-region Interior Reflective (STIR) approach (Branch et al., 1999) as implemented in the Python SciPy package (Virtanen et al., 2020), to solve the state parameters x iteratively. The method is based upon the Levenberg-Marquart method (Moré, 1978) and shows good stability for the boundary constraints.

## 2.1 Forward model

We used a vector radiative transfer model based on the successive order of scattering method for coupled atmosphere and ocean systems (Zhai et al., 2009, 2010) to model the measured reflectance and DoLP. The atmosphere is configured as three layers: a top molecular layer above the aircraft with only trace gas presented, a molecular layer below the aircraft in the middle, and an



aerosol and molecular mixing layer on the bottom (assumed uniformly distributed within 2 km from the ground) (Gao et al., 2019) as shown in the left panel of Fig. 1. The same vertical structure of the atmosphere was successfully used in the inversion of RSP data (Gao et al., 2019, 2020).

Aerosols are diverse in size, composition, and morphology. To capture their variation in the atmosphere, we modeled the size and refractive index for both fine and coarse modes. The aerosol size is represented by the volume density distribution as a combination of five lognormal distributions:

$$\frac{dV(r)}{dlnr} = \sum_{i=1}^{5} \frac{V_i}{\sqrt{2\pi}\sigma_{v,i}} \exp\left[-\frac{(\ln r - \ln r_{v,i})^2}{2\sigma_{v,i}^2}\right] \qquad (8)$$

where $V_i$ is the column volume density for each submode, the mean radius $r_i$ and standard deviation $\sigma_i$ are fixed with values of 0.1, 0.1732, 0.3, 1.0, 2.9 $\mu m$, and 0.35, 0.35, 0.35, 0.5, 0.5 respectively (Dubovik et al., 2006; Xu et al., 2016). The first three submodes are categorized as the fine mode aerosol, while the last two submodes are the coarse mode. All aerosols are assumed to be spherical in the current forward model. The nonspherical particle shape is important in the aerosol model (Dubovik et al., 2006), and will be considered in future studies. The aerosol refractive index spectra for the fine and coarse modes are represented by the principal component analysis in MAPOL (Wu et al., 2015; Gao et al., 2018) as

$$m(\lambda) = m_0 + \alpha_1 p_1(\lambda), \qquad (9)$$

where $p_1(\lambda)$ is the first-order principal component computed from the aerosol refractive index dataset including water, sea salt, dust-like particles, biomass burning, soot, sulfate, water-soluble, and industrial aerosols (Shettle and Fenn, 1979; d'Almeida et al., 1991). $m_0$ and $\alpha_1$ are two coefficients to determine the spectrum. For the application to AirHARP bands, $p_1(\lambda)$ for the real part of the refractive index is approximately spectral flat for both the fine and coarse mode aerosols within the visible spectrum. We further assume the spectral shape for the imaginary refractive spectra is also flat. Therefore, the two parameters can be combined into one to represent the refractive index. In this study, only four independent parameters are used to determine the real and imaginary refractive index spectra for the fine and coarse modes. With the aerosol size and refractive index, the polarimetric single scattering properties are modeled by the Lorenz-Mie theory and computed by the code developed by Mishchenko et al. (2002).





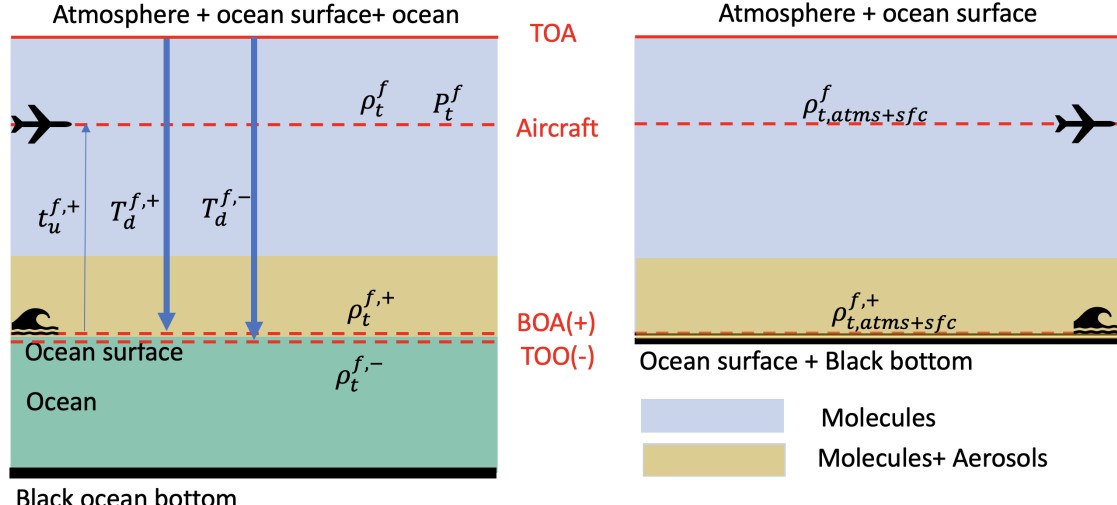

**Figure 1.** The left panel shows the coupled atmosphere and ocean system used in FastMAPOL including the atmosphere, ocean surface, and ocean body. The right panel represents a system used for atmospheric correction which only has atmosphere and ocean surface without scattering in the ocean body. The atmospheres in both systems are modeled as the same three layers. TOA indicates the top of the atmosphere. The bottom of the atmosphere (BOA) and the top of the ocean (TOO) indicate the locations just above and below the ocean surface, respectively. As discussed in the next section, all quantities shown in the figures need to be computed from the forward model and represented by the NN for efficient calculations. Symbols are defined in Table 1.

**Table 1.** Definition of the symbols for the quantities computed from the forward model (indicated by the superscript f) as shown in Fig. 1.

| Symbols | Definition |
|---------|-----------|
| $\rho_t^f$ | Reflectance at the aircraft level, Eq. (1) |
| $P_t^f$ | DoLP at the aircraft level, Eq. (2) |
| $\rho_t^{f,+}$ | Reflectance at BOA |
| $\rho_t^{f,-}$ | Reflectance at TOO |
| $\rho_{t,atms+sfc}^f$ | Reflectance at the aircraft level for atmosphere and ocean surface only |
| $\rho_{t,atms+sfc}^{f,+}$ | Reflectance at BOA for atmosphere and ocean surface only |
| $T_d^{f,+}$ | Irradiance transmittance from TOA to BOA, Eq. (12) |
| $T_d^{f,-}$ | Irradiance transmittance from TOA to TOO |
| $t_u^{f,+}$ | Radiance transmittance from BOA to sensor, Eq. (13) |

The molecular absorption properties are computed by a hyperspectral radiative transfer simulation (Zhai et al., 2009, 2010, 2018), including contributions from ozone, oxygen, water vapor, nitrogen dioxide, methane, and carbon dioxide under the US standard atmospheric constituent profiles (Anderson et al., 1986). Ozone is the most important gas that influences the absorption transmittance at the AirHARP bands of 550nm and 660nm. For the application to AirHARP measurements in ACEPOL, we





use the ozone column density as a free parameter with values from the Modern-Era Retrospective analysis for Research and Applications, Version 2 (MERRA-2) developed by NASA's Global Modeling and Assimilation Office (Gelaro et al., 2017) to rescale the molecular absorption optical depth calculated under the above-mentioned standard atmospheric profile.

For the ocean layer shown in Fig. 1, two ocean bio-optical models are implemented in the forward model of MAPOL: one with chlorophyll-a concentration (Chla; $mgm^{-3}$) as the single parameter applicable to open ocean optical properties, and the other with seven parameters more suitable to fully describe complex coastal waters (Gao et al., 2019). Since the waters are mostly clear within the ocean scenes in this study(Gao et al., 2020), open ocean model is used for both NN training and retrievals. The optical properties of open ocean waters include contributions from pure seawater, colored dissolved organic matter (CDOM), and phytoplankton, where the CDOM and phytoplankton absorption coefficients, and phytoplankton scattering coefficient and phase function are parameterized by Chla (Gao et al., 2019). Complex costal water model for NN trainings will be investigated in future studies. The ocean surface roughness is modeled by the isotropic Cox-Munk model with a scalar wind speed.

In summary, the parameters used to represent the forward model include five volume densities (one for each submode), four independent parameters for the refractive indices of fine and coarse modes, one parameter for wind speed, and Chla. Three additional geometric parameters are used to set up the system, including the solar zenith angle, viewing zenith angle, and relative viewing azimuth angle. Therefore, it requires a total of 15 parameters to conduct the radiative transfer calculation, with a total of 11 independent state parameters that can be retrieved from optimizing the cost function as defined in Eq. (3).

## 2.2 Remote sensing reflectance

An important task for the joint retrievals is to obtain the water leaving signal, which is often represented in ocean color studies by the spectral remote sensing reflectance defined as $R_{rs} = L_w^+/E_d^+$ where $E_d^+$ is the downwelling irradiance and $L_w^+$ is the water leaving radiance just above the ocean surface (Mobley et al., 2016). The remote sensing reflectance can be derived from the water leaving reflectance reaching the sensor ($\rho_w$) via:

$$R_{rs} = \left[\frac{\rho_w(\theta_0,\theta_v)}{\pi r^2}\right] \times \left[\frac{C_{BRDF}(\theta_0,\theta_v)}{T_d^{f,+}(\theta_0)t_u^{f,+}(\theta_0,\theta_v)}\right]. \tag{10}$$

where $\theta_0$ and $\theta_v$ are the solar and viewing zenith angles. $\rho_w$ represents the signals originating from scattering in the ocean that reached the sensor, which can be derived from the atmospheric correction process as

$$\rho_w(\theta_0,\theta_v) = \rho_t(\theta_0,\theta_v) - \rho_{t,atm+sfc}^f((\theta_0,\theta_v) \tag{11}$$

where $\rho_t$ is the measured total reflectance as defined in Eq. (1) and $\rho_{t,atms+sfc}^f$ is the reflectance from a system with only atmosphere and ocean surface (Mobley et al., 2016) as represented in the right panel of Fig. 1. The same formalism has been used to derive $R_{rs}$ from RSP measurements Gao et al. (2019, 2020).

The downwelling irradiance transmittance $T_d^f$ is for the solar irradiance from TOA to the surface, and the upwelling radiance transmittance $t_u^{f,+}$ is for the water leaving radiance from BOA to the sensor (Gao et al., 2019). Both $T_d^f$ and $t_u^{f,+}$ are denoted





in Fig. 1 and represented as follows:

$$T_d^{f,+}(\theta_0) = \frac{E_d^{f,+}(\theta_0)}{\mu_0 F_0} \tag{12}$$

$$t_u^{f,+}(\theta_0,\theta_v) = \left( \frac{\rho_t^f(\theta_0,\theta_v) - \rho_{t,atm+sfc}^f(\theta_0,\theta_v)}{\rho_t^{f,+}(\theta_0,\theta_v) - \rho_{t,atm+sfc}^{f,+}(\theta_0,\theta_v)} \right) \tag{13}$$

where $\rho_t^{f,+}$, $\rho_{t,atm+sfc}^{f,+}$ are reflectance just above ocean surface also denoted in Figure 1 and Table 1.

To remove the dependency of $R_{rs}$ on the solar and viewing geometries, a BRDF correction $C_{BRDF}$ is applied to adjust $R_{rs}$ to the observation with a zenith sun and a nadir viewing direction as defined by Morel et al. (2002):

$$C_{BRDF}(\theta_0,\theta_v) = \frac{\mathfrak{R}_o(W)}{\mathfrak{R}(\theta_v',W)} \times \frac{\rho_t^{f,-}(0,0)}{T_d^{f,-}(0)} \left[ \frac{\rho_t^{f,-}(\theta_0,\theta_v')}{T_d^{f,-}(\theta_0)} \right]^{-1} \tag{14}$$

where $\mathfrak{R}_o/\mathfrak{R}$ accounts for reflection and refraction effects when light propagates through the ocean interface. $C_{BRDF}$ in its original form is defined using the radiance and irradiance just below the ocean surface (Morel et al., 2002), here we have
converted all quantities into the radiance reflectance $\rho_t^{f,-}$ and the irradiance transmittance $T_d^{f,-}(\theta_0)$ similar to Eqs. (1) and (12). In our study, of AirHARP retrievals of remote sensing reflectance, we only consider the minimum viewing angle at each wavelength which is less than $1°$, the contribution of the $\mathfrak{R}$ factor is ignored due to its small angular variations (Morel and Berthon, 1989). All quantities denoted in Fig 1 need to be determined for the forward model and the calculation of remote sensing reflectance, and will be represented by NN models.

## 3    Neural network for forward model

Deep NN models are developing rapidly due to the advancement in machine learning infrastructure and demands in broad applications (Goodfellow et al., 2016), and are demonstrated to be efficient in approximating physical functions (Lin et al., 2017). In this study, we employed the deep feed-forward NN (Goodfellow et al., 2016) to represent the MAP measurements. In this section, we will discuss the procedures to train the NN forward model for AirHARP measurements, with its performance
evaluated.

### 3.1    Training data

To train a NN that can represent the forward model accurately for the AirHARP measurements from the ACEPOL field campaign, we generated the training data according to the average aircraft height of 20.1km on the day of 10/23/2017 from ACEPOL. We simulated 21,000 cases according to the forward model as discussed in the previous section by considering
general aerosol and ocean properties, as well as a large range of solar and viewing geometries with the minimum and maximum values of all parameters summarized in Table 2. The range of solar zenith angle $\theta_0$, viewing zenith angle $\theta_v$, and relative viewing azimuth angle $\phi_v$ are from $0°$ to $70°$, $60°$ and $180°$, respectively. The reflectance and DoLP with a viewing azimuth angle larger than $180°$ can be evaluated by the corresponding value less than $180°$ due to symmetry with respect to the principal plane (defined by $\phi_v = 0°$ and $\phi_v = 180°$). For each solar zenith angle, the polarized reflectance is calculated for all viewing





angles within the aforementioned ranges with an angular resolution of $1°$. The solar zenith angle, ozone column density, refractive index, and wind speed are randomly sampled in a linear scale. Chla is randomly sampled in a log scale. The fine mode volume fraction is sampled uniformly within [0, 1], which is then randomly partitioned to each submode. To maintain a uniform distribution of the total AOD, we sampled the AOD at 550nm within [0,0.5] in a linear scale. The volume density $V_i$

5 of each submode is determined by the total aerosol optical depth and volume fraction for each mode. Fig. 2 shows one example simulation dataset for the angular distribution of reflectance and DoLP.

**Table 2.** Parameters used to represent the atmosphere and ocean system for the radiative transfer simulation and NN training. $\theta_0$ and $\theta_v$ are the solar and viewing zenith angles. $\phi_v$ is the relative viewing azimuth angle. $V_i$ denote the five volume densities defined in Eq. (8). $m_r$ and $m_i$ are the real and imaginary parts of the refractive index. Ozone column density ($n_{O3}$) in the atmosphere, ocean surface wind speed, and Chla are also provided. The minimum (min) and maximum (max) values determine the parameter ranges used to generate NN training data, which are also the constraints in the retrieval algorithm. The initial values are the ones used in the retrieval optimization algorithm, where $\theta_0$, $\theta_v$, $\phi_v$ and $n_{O3}$ are assumed to be known from inputs.

| Parameters | Unit | min | max | initial |
|---|---|---|---|---|
| $\theta_0$ | Degree | 0 | 70 | (input) |
| $\theta_v$ | Degree | 0 | 60 | (input) |
| $\phi_v$ | Degree | 0 | 180 | (input) |
| $n_{O3}$ | Dobson | 150 | 450 | (input) |
| $m_r$(fine) | (None) | 1.3 | 1.7 | 1.5 |
| $m_r$(coarse) | (None) | 1.3 | 1.7 | 1.5 |
| $m_i$(fine) | (None) | 0 | 0.03 | 0.015 |
| $m_i$(coarse) | (None) | 0. | 0.03 | 0.015 |
| $V_1$ | $\mu m^3 \mu m^{-2}$ | 0 | 0.11 | 0.012 |
| $V_2$ | $\mu m^3 \mu m^{-2}$ | 0 | 0.05 | 0.007 |
| $V_3$ | $\mu m^3 \mu m^{-2}$ | 0 | 0.05 | 0.009 |
| $V_4$ | $\mu m^3 \mu m^{-2}$ | 0 | 0.19 | 0.017 |
| $V_5$ | $\mu m^3 \mu m^{-2}$ | 0 | 0.58 | 0.033 |
| Wind speed | m/s | 0.5 | 10 | 5.0 |
| Chla | $mg/m^3$ | 0.001 | 30 | 0.1 |





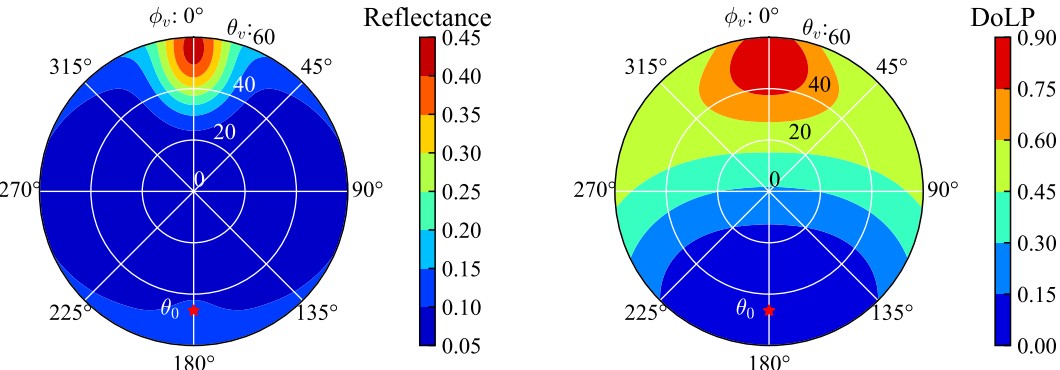

**Figure 2.** The reflectance (left panel) and DoLP (right panel) from radiative transfer simulation with the wind speed of 4.13 $m/s$, the aerosol optical depth of 0.26, Chla of 0.05 $mg/m^3$, and ozone column density of 196 Dobsons. The antisolar point is indicated by the red asterisk with a solar zenith angle $\theta_0 = 46.41°$. $\theta_v$ and $\phi_v$ indicate the viewing zenith and relative azimuth angles. The principal plane is defined by the viewing azimuth angle of $0°$ and $180°$.

We randomly selected 20,000 cases out of the total 21,000 simulated cases for the training and validation processes, and the remaining 1000 random cases will be used as test cases to evaluate the NN accuracy, which will be discussed in the next section. To enable the NN to predict reflectance and DoLP at any given viewing geometry, for each case, we sampled 100 random pairs of viewing zenith and azimuth angles. If the sampled angles fall outside of the pre-defined angular grids, values 5 from spline interpolation are used. The sunglint angles within an angle of $40°$ to the solar specular reflection direction are removed. Approximately 1 million data points are obtained for each wavelength for training.

### 3.2 Neural network training

A feed-forward NN can be defined recursively with one input layer, one output layer, and k hidden layers (Aggarwal, 2018):

$$\mathbf{h_1} = \Phi(\mathbf{W_1^T x + b_1}) \tag{15}$$

$$\mathbf{h_{p+1}} = \Phi(\mathbf{W_{p+1}^T h_p + b_{p+1}}), \mathbf{p = 1, ..., k - 1} \tag{16}$$

$$\mathbf{y} = \mathbf{W_{k+1}^T h_k + b_{k+1}} \tag{17}$$

where **x** is the input parameter vector including all 15 parameters needed to define the forward model as listed in Table 2. Here **x** not only contains the retrieval parameters in the state vector defined in Eq. (3) but also include additional non-retrieval parameters such as the solar zenith angle, viewing zenith and azimuth angles and the ozone column density. **y** is a four-15 dimensional output vector for reflectance or DoLP at the four AirHARP bands. The weight matrix $\mathbf{W_{p+1}}$ connects the p-th and (p+1)-th NN layers. The bias vector for the (p+1) layer is defined as $\mathbf{b_{p+1}}$. The output of each layer $\mathbf{h_{p+1}}$ becomes the input of the next layer as shown in Eq. 16. k is the number of hidden layers, k+1 refers to the output layer. In this study, we tested several NN architectures and eventually chose three hidden layers with the number of nodes of 1024, 256 and 128 as



shown in Table 3. The nonlinear activation function $\Phi$ used in this model is the LeakyReLU function. For each vector element, it is defined as

$$\Phi(z) = max(0, z) + 0.01 \times min(0, z). \tag{18}$$

The training process is to minimize the cost function defined as the mean square error between the training data generated from radiative transfer simulations and the NN predicted values (Aggarwal, 2018). All parameters in the neural network weight matrices and bias vectors, over 670,000 numbers, need to be trained. With this large number of parameters, it is a challenging task to avoid overfitting where the model works well for the training dataset but poorly for the dataset not used in the training process. Several training procedures are performed for reflectance and DoLP data to avoid overfitting and improve NN performance:

1. Both input and output data are normalized before training. We normalize the input data into the range of [0,1] using the minimum and maximum values from the datasets as listed in Table 2. The reflectance and DoLP in the output layers are normalized by dividing their standard deviation of the training data at each wavelength.

2. The Adaptive Momentum (Adam) algorithm (Kingma and Ba, 2014) with weight decay regularization (Loshchilov and Hutter, 2019) is used to update the weights and bias of the NN. The training dataset is divided into multiple mini-batches, each with 1024 random samples. The training iterations loop through all mini-batches in the training data (each loop is called an epoch). Convergence requires training through multiple epochs, where mini-batches are resampled in each epoch.

3. The learning rate determines the step size in the parameter update. We use an exponential decay schedule to reduce the learning rate: we start with a learning rate of 0.005 and reduce the learning rate by a factor of 10 every 200 epoches.

4. To monitor over-fitting in the training process, we split the data into 70% for training and 30% for validation. We conduct the optimization based on the training dataset, and meanwhile, we monitor the performance of training by applying the NN model on the validation dataset. To avoid overfitting, the early-stopping approach is employed where the training is stopped when the cost function on the validation dataset stops to reduce for a threshold of 50 epochs.

The machine learning Python library Pytorch is used for the training (Paszke et al., 2019). The trained NN model is used to replace the radiative transfer model to compute the reflectance and DoLP in the retrieval algorithm. The Jacobian matrix used in the optimization is computed by the finite difference approximation of the partial derivatives of reflectance and DoLP with respect to the retrieval parameters. Here central difference method is used. Note that the Jacobian matrix can also be computed analytically from the NN model using the automatic differentiation techniques based on the chain rule of differentiation (Baydin et al., 2018). This will be a topic in our future studies.

## 3.3 Neural network accuracy

After training the NN model, we evaluated its accuracy using synthetic AirHARP measurements generated from the 1000 simulation cases which have not been used in the training and validation process. Each simulation dataset includes polarized



reflectance on regular viewing angle grids, which are interpolated to the viewing geometry of AirHARP to create synthetic measurement data and compare with the NN predictions. Glint angles are excluded from the comparison because the NNs are not trained over these angles. As one example shown in Fig. 4, both the reflectance and DoLP are in good agreement between the synthetic data and the NN results, where the maximum absolute differences for reflectance and DoLP are within 0.001 and

5   0.0025. This translates to a difference for both reflectance and DoLP mostly less than 1%. The maximum percentage difference can be as large as 3% for 870nm bands due to the small reflectance magnitude.

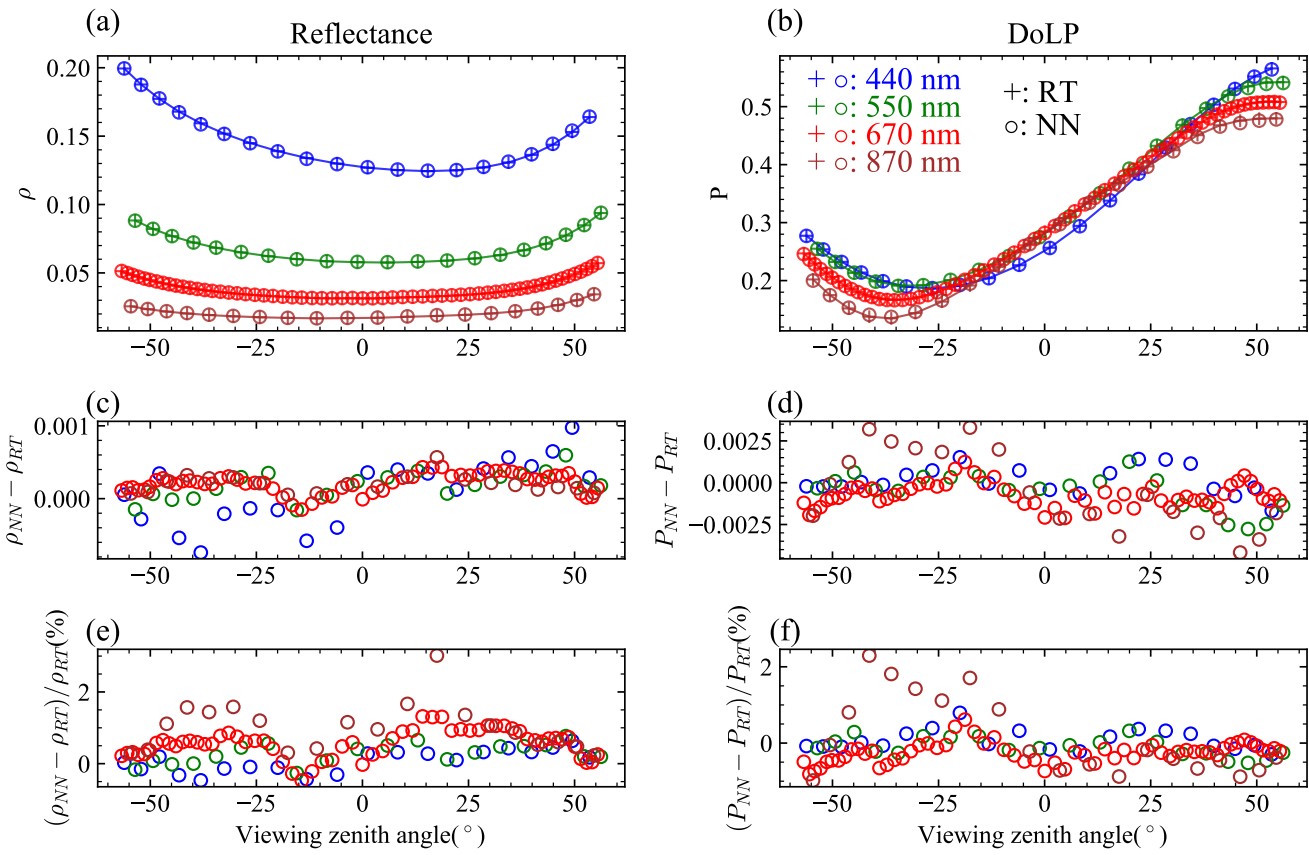

**Figure 3.** The synthetic HARP reflectance (left panel) and DoLP (right panel) sampled from the radiative transfer data shown in Fig. 2. The positive and negative signs of the viewing zenith angles indicate the azimuth angles of $\phi_v = 116.2°$ and $180° + \phi_v$.

The comparison with all 1000 synthetic datasets and their NN predictions are shown in Fig. 4. The mean absolute error (MAE), and the root mean square error (RMSE) between the simulation data ($T_i$) and the NN predicted data ($R_i$) shown in





Fig. 4 are defined as

$$\text{MAE} = \frac{1}{N} \sum_{i=1}^{N} |R_i - T_i|, \tag{19}$$

$$\text{RMSE} = \sqrt{\frac{1}{N} \sum_{i=1}^{N} (R_i - T_i)^2}. \tag{20}$$

Both MAE and RMSE are useful metrics, where MAE has less dependency on outliers comparing with RMSE.

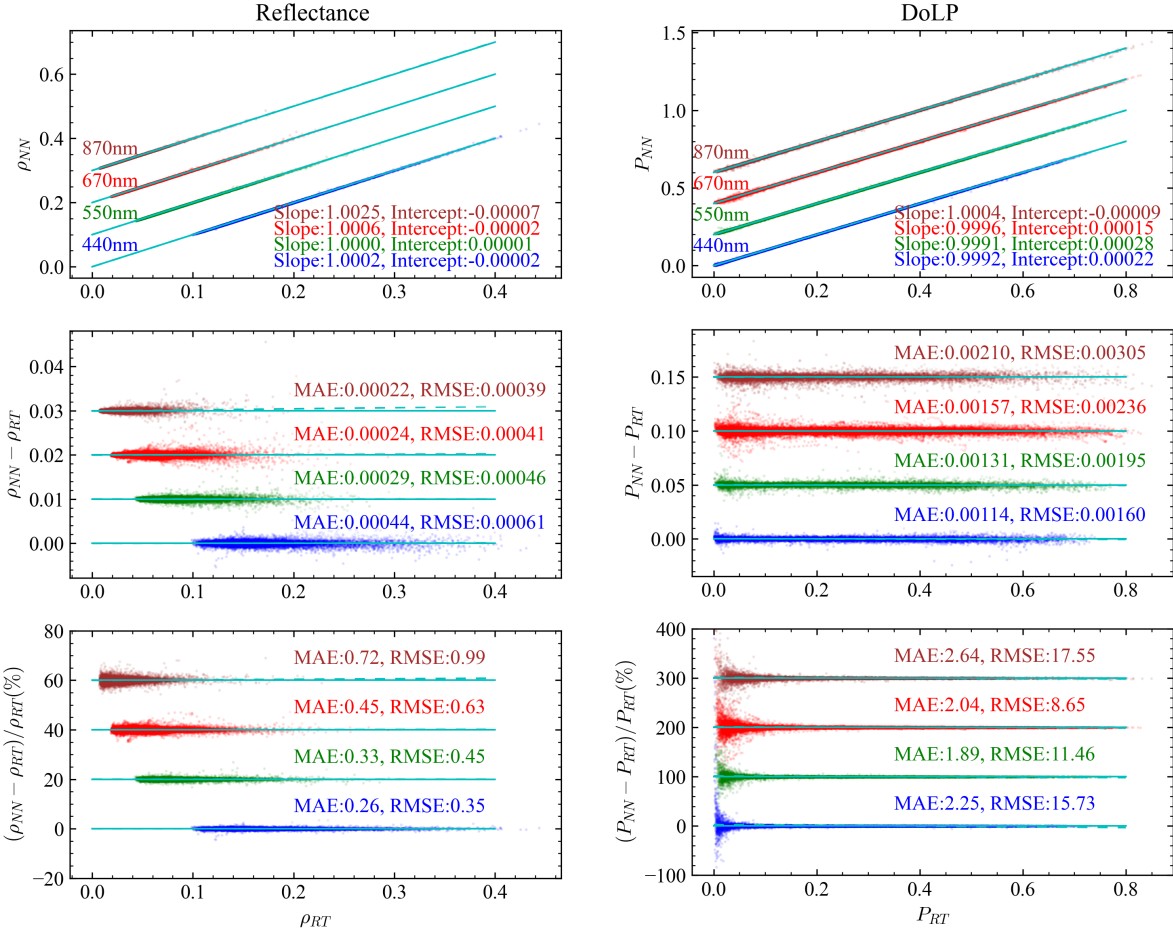

**Figure 4.** Comparison between the radiative transfer simulation and NN prediction, left panel: reflectance ($\rho$); right panel: DoLP ($P$). The scatter plots are shown in the top panel, the absolute different in the middle panel, and the percentage difference in the bottom panel. For each plot, the data points for the 550, 660 and 870nm bands are shifted upward by constant offsets consecutively as indicated by the solid cyan lines.



Analysis shows that the statistics of the differences between the NN prediction and the RT simulations as shown in Fig. 4 can be well modeled by Gaussian distributions and characterized by RMSE. Therefore the RMSE is used to represent the NN uncertainties for both reflectance ($\sigma_{\rho,NN}$) and DoLP ($\sigma_{\rho,NN}$) and will be incorporated into the total uncertainties in the cost function. Table. 3 summarizes the uncertainties of the NN models. The $\sigma_{\rho,NN}$ at 440 nm is 0.0006, which decreases to

0.0004 at 870 nm. However, due to the smaller reflectance magnitude at 870nm, the corresponding RMSE for the percentage reflectance difference as shown in Fig. 4 is increased from 0.4% at 440nm to 1.0% at 870nm. For DoLP, the maximum $\sigma_{P,NN}$ is 0.003 at 870 nm which decreases to 0.0016 at 440 nm. The uncertainties can be further improved with more training data points. For the readers' information, RMSE of the NN model trained with 20,000 cases (1 million data points) decreases by a factor of $\sqrt{2}$ in comparison with the one using 10,000 cases (0.5 million data points). It takes 0.01s in a single-core CPU

(AMD EPYC Processor) or 1 ms in a GPU (GeForce GTX 1060) to predict all 120 angles for both reflectance and DoLP in the NN forward model.

**Table 3.** The accuracy of the NN for the corresponding quantities in terms of the RMSE ($\sigma_{NN}$) of the difference between the NN predicted values and the truth values from radiative transfer simulation. The NN architecture denotes the number of the nodes in each layer. Remote sensing reflectance is computed by Eq.(10) using the NNs for $\rho^f_{t,atms+sfc}$ and $[C_{BRDF}/T_d t_u]$ as discussed in Section 3.4.(The percentage values listed below in the parenthesis are the percentage uncertainties defined as the RMSE of the percentage difference between the RT simulation and NN predictions.)

| Quantities | NN architecture | $\sigma_{NN}(440nm)$ | $\sigma_{NN}(550nm)$ | $\sigma_{NN}(670nm)$ | $\sigma_{NN}(870nm)$ |
|:---:|:---:|:---:|:---:|:---:|:---:|
| $P^f_t$ | $15 \times 1024 \times 256 \times 128 \times 4$ | 0.0016 | 0.0020 | 0.0024 | 0.0030 |
| $\rho^f_t$ | $15 \times 1024 \times 256 \times 128 \times 4$ | 0.00061(0.4%) | 0.00046(0.5%) | 0.00041 (0.6%) | 0.00039 (1.0%) |
| $\rho^f_{t,atms+sfc}$ | $14 \times 1024 \times 256 \times 128 \times 4$ | 0.00084(0.4%) | 0.00065(0.6%) | 0.00057 (0.9%) | 0.00055 (1.3%) |
| $\left[\frac{C_{BRDF}}{T_d t_u}\right]$ | $15 \times 128 \times 128 \times 4$ | 0.02(0.9%) | 0.01(0.7%) | 0.01(1.0%) | 0.01(1.0%) |
| $R_{rs}$ | *Eq (10) | 0.0004 | 0.0002 | 0.0002 | 0.0001 |

The assessment of the NN accuracy is relative to the synthetic measurements simulated by the vector radiative transfer simulations. To account for the modeling uncertainties of the forward model $\sigma_f$, we consider both the NN accuracy $\sigma_{NN}$ and the numerical accuracy of the radiative transfer simulations $\sigma_{RT}$ for reflectance and DoLP, respectively. Uncertainties

due to incomplete assumptions in the forward model are not considered. Several internal parameters determine the numerical accuracy of the radiative transfer simulations. In the framework of the successive order of scattering (Zhai et al., 2008, 2009), these parameters include the number of scattering orders (Ns), the number of Gaussian quadratures for discretizing the viewing zenith angle in the atmosphere ($P_a$) and ocean ($P_o$), and the order of Fourier decomposition ($M$) for the viewing azimuth angle, and the order of Legendre expansion ($L$) of the single scattering phase function. In this study, we chose $N = 20$,

$P_a = 32$, $P_o = 64$, $M = 32$, and $L = 32$, which has a higher accuracy than the radiative transfer forward model directly used in our previous retrieval studies Gao et al. (2020).



To quantify the accuracy of the radiative transfer calculation used for generating training data ($\sigma_{RT}$), we simulated an additional 1000 synthetic AirHARP datasets with all internal parameters doubled as the most rigorous calculations, and the viewing angular resolution was reduced from $1°$ to $0.5°$ in order to reduce interpolation errors. The resultant reflectance and DoLP values are compared between these two sets of radiative transfer calculations. The RMSE for each band can be used as

a measure of the accuracy for the radiative transfer calculation used to generate the training data ($\sigma_{RT}$). The uncertainties $\sigma_{RT}$ for reflectance and DoLP are summarized in Table 4, with reflectance uncertainties less than 0.00015 and DoLP uncertainties less than 0.0007 for all AirHARP bands. $\sigma_{\rho,RT}$ is about four times smaller than the NN uncertainties, and $\sigma_{P,RT}$ is about 4 to 10 times smaller. Therefore, NN uncertainties are not dominated by the uncertainties of the RT simulation. The measurement uncertainties from calibration ($\sigma_{cal}$) and pixel averaging ($\sigma_{avg}$) as discussed in Section 2 are also summarized in Table 4,

which shows the total forward model uncertainties $\sigma_f = \sqrt{\sigma_{RT}^2 + \sigma_{NN}^2}$ as approximated in Eq. (7) are much smaller than the total measurement uncertainties $\sigma_m = \sqrt{\sigma_{cal}^2 + \sigma_{avg}^2}$ as defined in Eq. (6). The overall uncertainties used in the retrieval cost function in Eq (3) are dominated by the measurement contributions as we expected.

**Table 4.** Comparisons of the uncertainties for reflectance ($\rho$) and DoLP (P) for both measurement and forward model including calibration uncertainty ($\sigma_{cal}$), the uncertainty from averaging AirHARP pixels into a 550 m $\times$ 550 m box ($\sigma_{avg}$), the radiative transfer simulation uncertainty ($\sigma_{RT}$), and the NN uncertainty ($\sigma_{NN}$). Same $\sigma_{\rho,NN}$ and $\sigma_{P,NN}$ have been shown in Table 3, and are repeated here for comparisons. (Same as Table 3, the percentage values listed in the table indicate the percentage uncertainties.)

| Uncertainties | 440nm | 550nm | 670nm | 870nm |
|---|---|---|---|---|
| $\sigma_{\rho,cal}$ | 3% | 3% | 3% | 3% |
| $\sigma_{\rho,avg}$ | 0.0006 (0.3%) | 0.0004 (0.5%) | 0.0004 (1.0%) | 0.0004 (1.7%) |
| $\sigma_{\rho,RT}$ | 0.00012 (0.08%) | 0.00005 (0.07%) | 0.00010 (0.2%) | 0.00015 (0.4%) |
| $\sigma_{\rho,NN}$ | 0.00061 (0.4%) | 0.00046 (0.5%) | 0.00041 (0.6%) | 0.00039 (1.0%) |
| $\sigma_{P,cal}$ | 0.01 | 0.01 | 0.01 | 0.01 |
| $\sigma_{P,avg}$ | 0.006 | 0.006 | 0.009 | 0.020 |
| $\sigma_{P,RT}$ | 0.0002 | 0.0002 | 0.0005 | 0.0007 |
| $\sigma_{P,NN}$ | 0.0016 | 0.0020 | 0.0024 | 0.0030 |

Furthermore, in this study higher accuracies from the radiative transfer simulations are used for the NN training for comparison with the accuracies from the radiative transfer model directly used in our previous retrieval algorithm. Since the simulations

of the training data can be conducted independent to the retrieval algorithm, higher computational costs can be accommodated to improve NN forward model accuracy. After the NN model is trained, the model can be applied to the retrieval algorithm through efficient matrix operations.



### 3.4 Neural network model for remote sensing reflectance

As discussed in Sect. 2.2, the water leaving signals are represented by the remote sensing reflectance as defined in Eq. (10) (Mobley et al., 2016). To conduct the atmospheric correction in Eq. (11), we need to determine the reflectance $\rho^f_{t,atmos+sfc}$ at the aircraft level, transmittance $t^f_u$ and $T^f_d$, and the BRDF correction coefficient $C_{BRDF}$. Based on Eq. 10, we combined $T^{f,+}_d$,

$t^{f,+}_u$, and $C_{BRDF}$ into a single term denoted as $[C_{BRDF}/T_d t_u]$. To efficiently determine $R_{rs}$, two NNs need to be trained to represent $\rho^f_{t,atmos+sfc}$ and $[C_{BRDF}/T_d t_u]$, respectively,

    Following similar NN training schemes as discussed previously, we conducted 10,000 simulations to determine the reflectance at aircraft altitude $\rho_{t,atmos+sfc}$ from a system with only atmosphere and ocean surface (right panel of Fig. 1), and trained the NN for $\rho_{t,atmos+sfc}$ in the same way as $\rho^f_t$. Since this system only includes atmosphere and ocean surface but

without ocean body, there are total 14 input parameters (without Chla). To train a NN for $[C_{BRDF}/T_d t_u]$ with $T^{f,+}_d$, $t^{f,+}_u$, and $C_{BRDF}$ defined in Eqs. (12),(13) and (14), we obtained five additional quantities corresponding to the above-mentioned 10,000 cases with and without ocean body: for the fully coupled system with atmosphere, ocean surface and ocean body (left panel of Fig. 1), we computed the reflectance just above and below the ocean surface ($\rho^{f,+}_t$ and $\rho^{f,-}_t$), and irradiance transmittance just above and below the ocean surface ($T^{f,+}_d$ and $T^{f,0}_d$); for the system without ocean body but with ocean surface (right panel of

Fig. 1), we computed the reflectance just above the ocean surface ($\rho^{f,+}_{t,atms+sfc}$). The accuracy of the NNs for $\rho^f_{t,atmos+sfc}$ and $[C_{BRDF}/t_u T_d]$ are evaluated and shown in Table 3, which are in the same order of uncertainty magnitudes in percentage as other quantities.

    To evaluate the overall accuracy for the $R_{rs}$ after the BRDF correction, we conducted radiative transfer simulations with a zenith sun and a nadir viewing direction, and obtained the truth remote sensing reflectance using the upwelling radiance and

downwelling irradiance just above the ocean surface as examples shown in Fig. 5. The predicted $R_{rs}$ were computed from Eq. (10) after the application of two NNs. The RMSE of the difference between the simulated and NN predicted $R_{rs}$ are shown in Table 3 with a maximum value of 0.0004 at wavelength 440 nm, and smaller than 0.0002 in other bands. These values including all uncertainties including the accuracy of the NN and assumptions in the BRDF model.





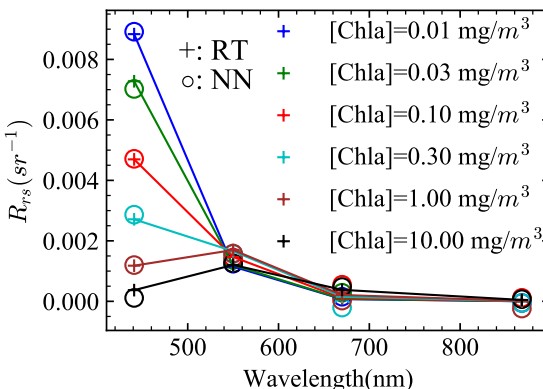

**Figure 5.** Comparison of the truth $R_{rs}$ (RT) and the neural network (NN) computed $R_{rs}$. The truth $R_{rs}$ is computed from radiative transfer simulations with a zenith sun and nadir viewing direction. The NN computed $R_{rs}$ is following Eq. (10).

## 4 Joint retrieval results on synthetic AirHARP measurements

The NN forward models for reflectance ($\rho_t^f$) and DoLP ($P_t^f$) are used in the FastMAPOL retrieval algorithm as discussed in Sect. 2. To evaluate the performance of the retrieval algorithm, we conducted retrievals on the synthetic AirHARP data. The creation of the synthetic data is discussed in Sect. 3.3. To account for the measurement uncertainties, random noise is added to the simulated data according to the calibration uncertainties as listed in Table 4. The total uncertainties in the cost function include contributions from calibration ($\sigma_{cal}$), radiative transfer simulation ($\sigma_{RT}$), and NN model ($\sigma_{NN}$). Uncertainties from pixel averaging ($\sigma_{avg}$) for the AirHARP field measurements are not considered in the synthetic dataset.

Using the initial values as listed in Table 2, a total of 1000 synthetic AirHARP cases are retrieved with the cost function values($\chi^2$) summarized in Fig 6. Retrievals with $\chi^2 < 1.5$ are chosen in our following discussion, which includes 96% of all retrieval cases. Gao et al. (2020) showed that the retrieval results depend on the initial values. Testing with several random sets of initial values shows that the statistics of the retrieval results from the 1000 synthetic cases are robust. As demonstrated by Di Noia et al. (2015) and Di Noia et al. (2017), a better choice of initial values for each pixel in the optimization may further improve the overall retrieval accuracy.





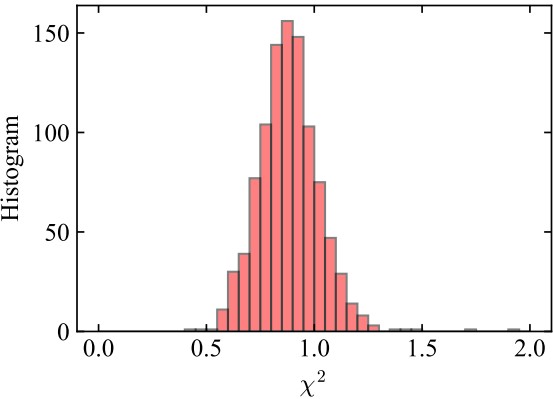

**Figure 6.** Histogram of the cost function values ($\chi^2$) with initial values as specified in Table 3 with a total of 1000 cases. The most probable $\chi^2$ is 0.82. A threshold of $\chi^2 < 1.5$ is used in the discussion.

With the directly retrieved aerosol refractive index and volume densities (see Table 2) as inputs, the aerosol optical depth (AOD) and single scattering albedo (SSA) for both the fine and coarse modes, were computed using additional NNs to represent the Lorenz-Mie calculations in Appendix A. The retrieved total AOD, SSA, wind speed, and Chla are compared with the truth values as shown in Fig. 7. Total AOD indicates the summation of the fine and coarse mode AODs and total SSA is the ratio

of the total scattering and extinction cross-sections, both are specified in in Appendix A. For fine aerosol, the AOD, SSA, refractive index ($m_r$), effective radius ($r_{eff}$) and variance ($v_{eff}$) are shown in Fig. 8. The color plots indicate the data point density (normalized by its maximum value) approximated by a kernel density estimation method (Silverman, 1986).

In order to quantify the variation of the retrieval uncertainties with respect to different aerosol loadings, we computed the the RMSE between the retrieved and truth values at five AOD ranges including [0.01,0.1], [0.1,0.2], [0.2,0.3],[0.3,0.4] and

[0.4,0.5]. The each AOD ranges includes an approximate 200 cases. Note that as discussed in Sect. 3.3, the total AOD and the fine mode volume fraction are uniformly sampled for the simulated data, therefore, there is an equal mixing fraction of fine and coarse mode aerosol for each AOD range. The retrieval uncertainties for aerosols are shown in Figure 9 with the corresponding ranges indicated by AOD values from 0.1 to 0.5. All discussions regarding the AOD and SSA are for wavelength of 550nm in this section.

As shown in Fig. 7 and Fig. 9, the errors of the retrieved total AOD increase with aerosol loadings: the uncertainty (evaluated using RMSE) is 0.008 and 0.015 for the AOD range [0.01, 0.1] and [0.1,0.2], and increases to 0.035 for the AOD range [0.4,0.5]. Similar absolute uncertainties are found for both the fine and coarse mode AODs with a value slightly smaller. In percentage, the total AOD uncertainties is 28.3% at the AOD range [0.01, 0.1] where the large uncertainties is due to the cases with small AODs. For the AOD range from [0.1,0.2] to [0.4,0.5], the AOD uncertainties further decrease from 14.4% to 5.6%.

Similar to the total AOD uncertainties, the total SSA uncertainties decreases with AOD from 0.05 to 0.02. The fine mode SSA uncertainties reduce similarly from 0.05 to 0.03. The uncertainties for coarse mode SSA reduces slightly from 0.1 to 0.08



which is more than twice larger than the fine mode SSA uncertainties. The uncertainties for the fine mode $m_r$, $r_{eff}$ and $v_{eff}$ shows a larger value in the AOD bin of [0.01,0.1] of 0.06, 0.024 $\mu m$, 0.08, and then remain close to constant with a value of 0.03, 0.01 $\mu m$ and 0.03 respectively. The averaged uncertainties for coarse mode $m_r$, $r_{eff}$ and $v_{eff}$ are approximately 0.08, 0.5 $\mu m$ and 0.15 respectively weakly AOD dependency. The coarse mode $m_r$ uncertainty are more than twice to the fine mode

5  uncertainty. The larger uncertainty values for coarse mode $r_{eff}$ and $v_{eff}$ are also related to their large particle size.

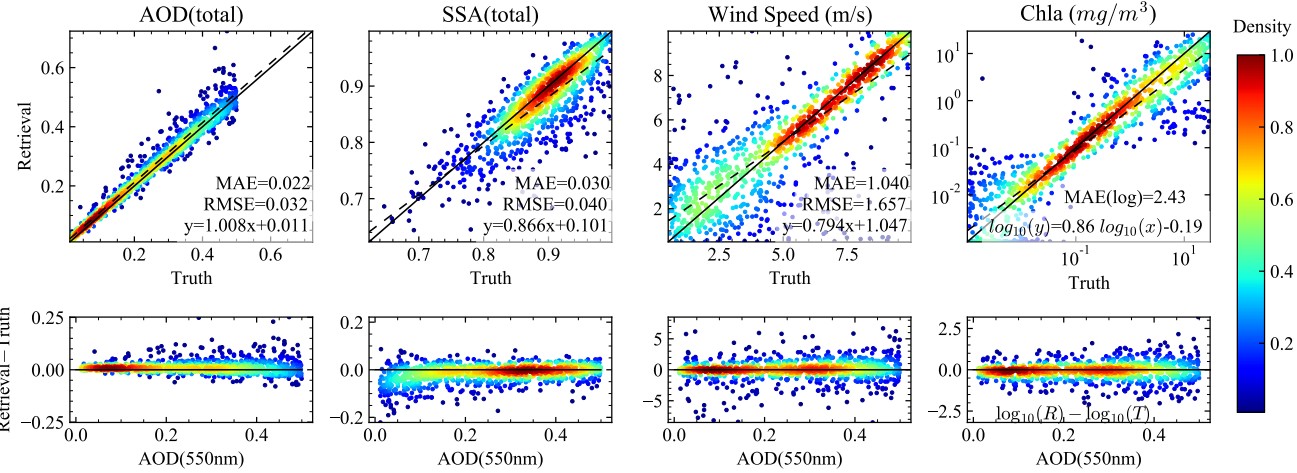

**Figure 7.** The comparisons of the retrieved and truth values for total AOD (550 nm), SSA (550 nm), wind speed, and Chla are shown in the top panels. The dashed line indicates the linear regression fitting with $y = \beta x + \alpha$, where $\beta$ is the slope and $\alpha$ is the intercept. The lower panels show the difference between the retrieved and truth values of the corresponding upper panel parameters as a function of the total AOD at 550 nm.

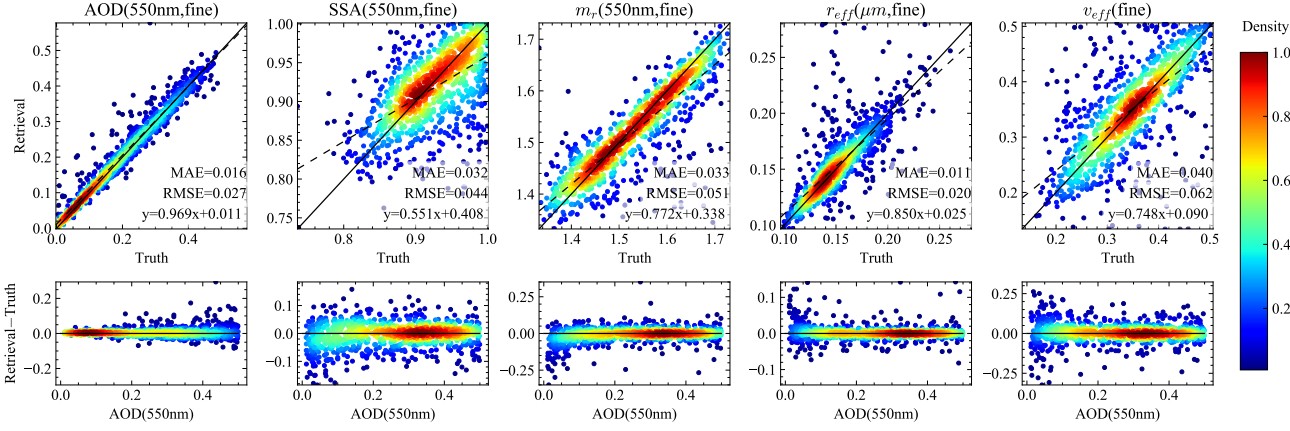

**Figure 8.** The comparisons of the retrieved and truth values for the fine mode aerosol parameters including AOD, SSA, refractive index ($m_r$), effective radius ($r_{eff}$) and variance ($v_{eff}$).





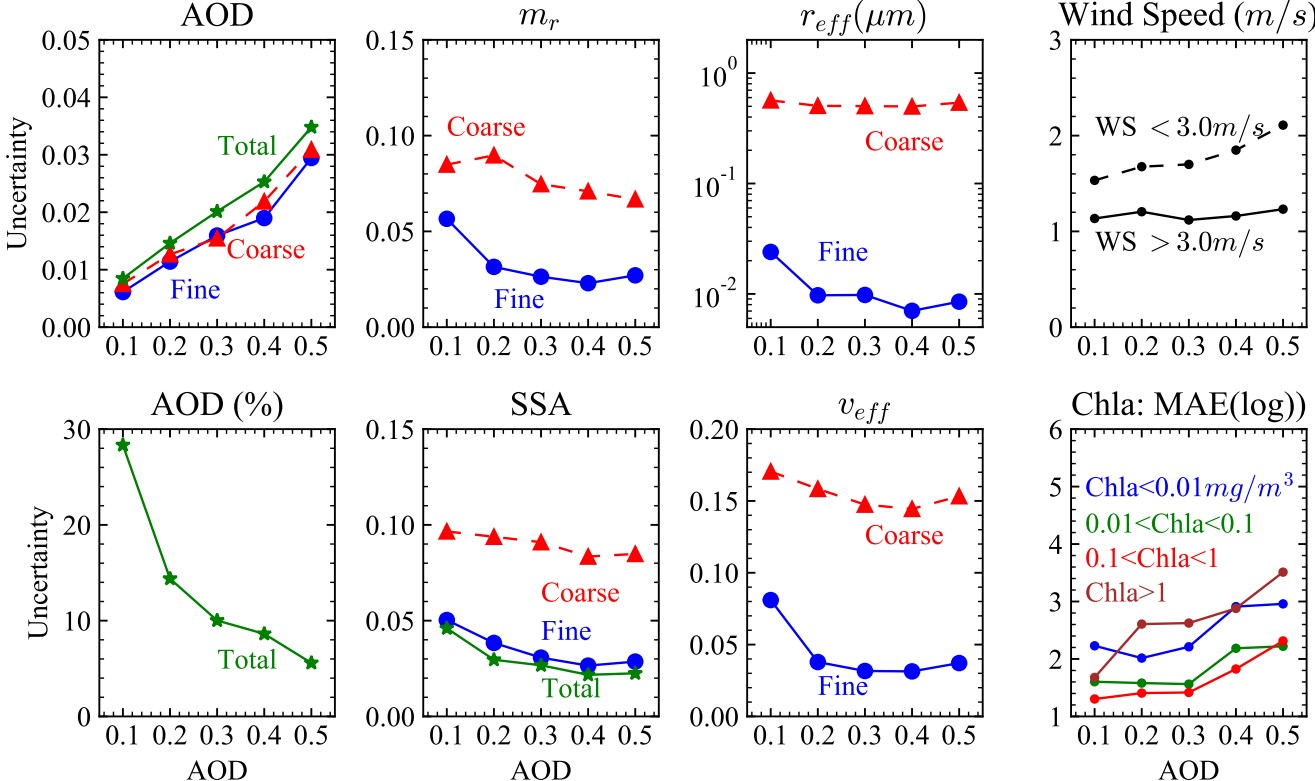

**Figure 9.** The retrieval uncertainties at various aerosol loadings for AOD, SSA, refractive index ($m_r$), effective radius ($r_{eff}$) and variance ($v_{eff}$), wind speed, and Chla. AOD values at the x-axis from 0.1 to 0.5 indicate the five ranges of total AOD including [0.01,0.1], [0.1,0.2], [0.2,0.3],[0.3,0.4] and [0.4,0.5] which are used to compute the corresponding uncertainties. Chla uncertainties are evaluated in terms of MAE in log scale (see Eq. (21)) and all other parameters are evaluated in terms of RMSE. AOD(%) indicate the percentage AOD uncertainties comparing to the truth AOD.

For wind speed retrievals as shown in Fig. 7 the agreements between the truth and retrievals depend strongly on the wind speed value itself: when the wind speed is small, there is less retrieval sensitivity due to the removal of glint; for larger wind speed, the agreements are improved, likely due to the larger range of angles influenced by wind speed. The retrieval uncertainties are shown in Fig. 9, for wind speed (WS) smaller than 3 m/s, the uncertainty increases from 1.5 to 2.1 m/s for AOD ranges from [0.1,0.2] to [0.4, 0.5]. While for wind speed larger than 3 m/s, the retrieval uncertainty is almost an constant of 1.2 m/s with a small increase less than 0.1 m/s. The retrieved and truth Chla is compared in Fig. 7 where the MAE in log scale, or MAE(log), is used with definition as:

$$\text{MAE(log)} \quad = \quad 10^Y \text{ where } Y = \frac{1}{N} \sum_{i=1}^{N} |\log_{10}(R_i) - \log_{10}(T_i)| \tag{21}$$





where $R_i$ and $T_i$ denote the retrieval and truth values. MAE(log) is recommended by Seegers et al. (2018) as a better metrics for Chla, which indicates the averaged ratio between the retrieval and truth values. The dependency of the MAE(log) for Chla with the aerosol loadings is shown in Fig. 9. The Chla retrieval performance depends on the magnitude of the Chla. In this work, we chose four ranges of Chla according to the trophic regions discussed in (Seegers et al., 2018). Note that

Chla from in-situ measurements is typically lager than $0.01mg/m^3$ and we chose a broader range of Chla with its minimum value of $0.001mg/m^3$ as listed in Table 2 for sensitivity studies. For $0.01mg/m^3 < $ Chla $< 0.1mg/m^3$ and $0.1mg/m^3 < $ Chla $< 1mg/m^3$, Chla retrieval uncertainties vary within 1.3 to 1.6 when AOD<0.3, and then increase to 2.3 at AOD range [0.4,0.5]. For Chla $> 1mg/m^3$ and Chla $< 0.01mg/m^3$, the uncertainties are generally larger with a value around 2 to 3.

With the retrieved aerosol and ocean properties, the atmospheric correction procedures can be applied to compute the remote

sensing reflectance as discussed in Section 3.4. The comparison of the retrieved $R_{rs}$ with the truth data is shown in Fig. 5. To account for the various solar geometries, the BRDF correction has been applied on the retrieved $R_{rs}$ as discussed in Section 3.4. The truth $R_{rs}$ was computed with a zenith sun and a nadir viewing direction, emphasizing the need for the latter correction to the MAP observations. Overall $R_{rs}$ uncertainties for the four bands are 0.007, 0.0004, 0.0002 and 0.0002 as shown by the RMSE in Fig. 10. MAE showed values of 0.0006, 0.0003, 0.0002 and 0.0001, which are less sensitive to the outliers. Note

that the atmospheric correction is applied on the synthetic measurements without adding additional random noise in order to evaluate the impacts on $R_{rs}$ uncertainties from only aerosol and ocean surface properties retrievals. The retrieval uncertainties for $R_{rs}$ for each AirHARP bands are shown in Fig. 10 depending on the aerosol loadings: larger uncertainties are found with larger aerosol optical depth.

The PACE accuracy requirements on ocean color are specified in terms of the water-leaving reflectance, which can be

converted to those of $R_{rs}$ by dividing them by a factor of $\pi$. The resultant requirements in terms of $R_{rs}$ are 0.0006 $sr^{-1}$ or 5% from 400 to 600 nm, and 0.0002 $sr^{-1}$ or 10% from 600 to 710 nm (Werdell et al., 2019). As shown in Fig. 11, $R_{rs}$ at 550nm are within the requirement of 0.0006 $sr^{-1}$ for all AOD ranges. For 440 nm band, the $R_{rs}$ retrieval uncertainties are larger than the requirement when AOD is larger than 0.3. $R_{rs}$ at 670 and 870 nm varies in a very small dynamical range and has less impacts by the aerosol retrievals. $R_{rs}$ uncertainties at 670 and 870 nm are slightly larger than the requirement of 0.0002 $sr^{-1}$

when AOD(550nm) is larger than 0.4 and 0.3 respectively. Further work is needed to understand how the uncertainties of the retrieved aerosol properties influence the retrievals. Note that from Table 3, the uncertainties of the $R_{rs}$ computed using NNs have an uncertainty of 0.0004 to 0.0001 from 440nm to 870nm, which may be further reduced with better training and help the reduction of the $R_{rs}$ retrieval uncertainties.





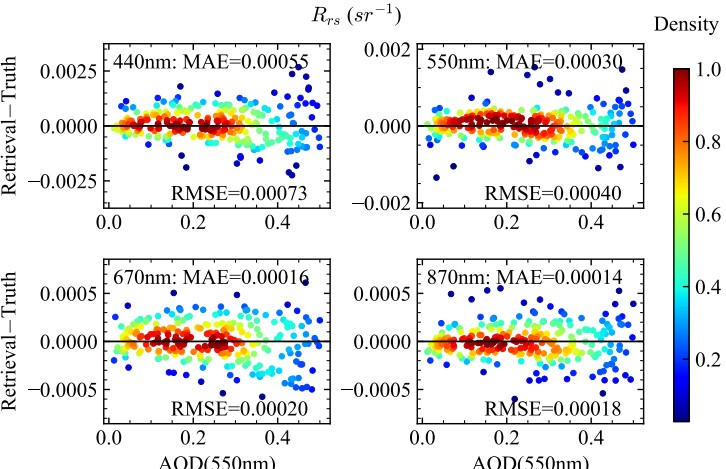

**Figure 10.** The difference between the retrieval and truth $R_{rs}$ with respect to AOD. The truth $R_{rs}$ is computed with a zenith sun and a nadir viewing direction. The retrieved $R_{rs}$ is following Eq.(10) with the BRDF correction considered. RMSE and MAE are for all retrievals cases at each wavelength.

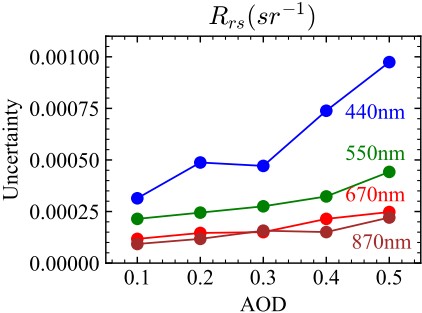

**Figure 11.** Retrieval uncertainties for $R_{rs}$ at the four AirHARP bands. The uncertainties are computed in the same way as for Fig.9 in terms of RMSE.

# 5    Joint retrieval results on AirHARP measurements from ACEPOL

The ACEPOL field campaign, conducted from October to November of 2017, included a total of six passive and active instruments on the NASA ER2 high-altitude aircraft (Knobelspiesse et al., 2020) with four MAPs: AirHARP (McBride et al., 2020)), AirMSPI (Diner et al., 2013), SPEX Airborne (Smit et al., 2019) and the RSP (Cairns et al., 1999), and two lidars: HSRL-2 (Burton et al., 2015) and CPL (the Cloud Physics Lidar) (McGill et al., 2002). Aerosol retrieval algorithms have been applied for all four MAPs (Fu et al., 2020; Puthukkudy et al., 2020; Gao et al., 2020). The measurement datasets are available from the ACEPOL data portal (Knobelspiesse et al., 2020). In this work, we focus on the study of the AirHARP measurements over





ocean scenes as shown in Fig. 12 on Oct 23, 2017. The viewing angles are within $\pm 57°$ along-track, and $\pm 47°$ cross-track as shown in the polar plots in Fig. 12. Fig. 13 shows the RGB images (670, 550 and 440nm) for the three scenes at near nadir viewing direction.

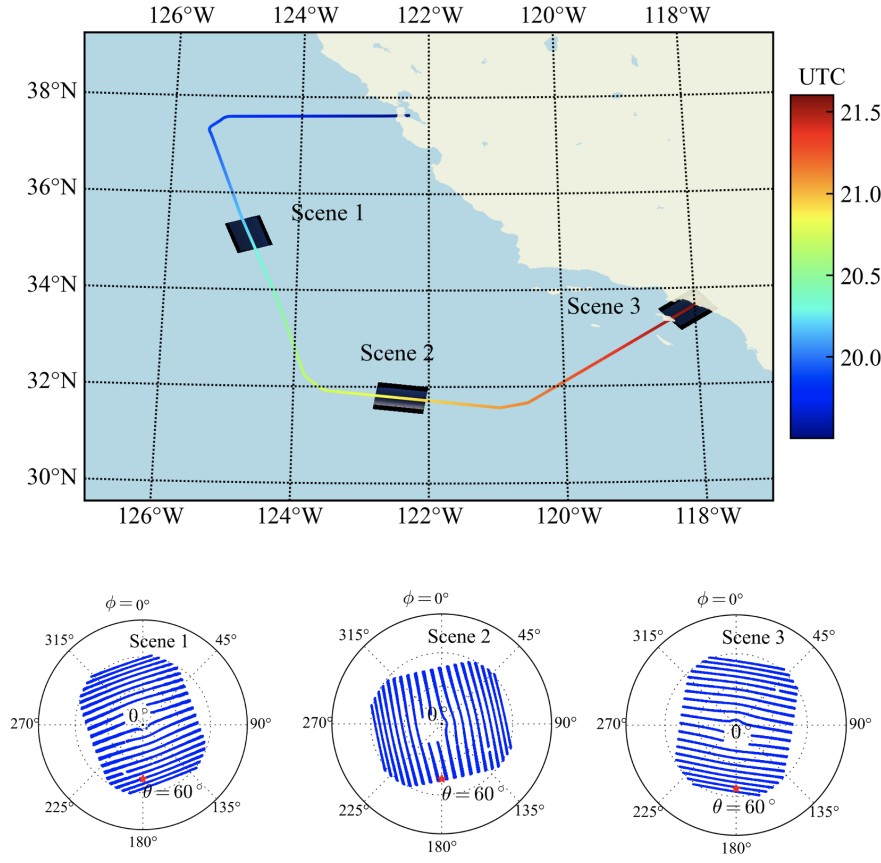

**Figure 12.** The location of the three ocean scenes from AirHARP from ACEPOL on Oct 23, 2017. The flight track is color labeled by the UTC. The aircraft flew at an altitude of 20.1 km. The viewing zenith and relative azimuth angle (relative to the solar azimuth angle) for the 440 nm band from all pixels in the corresponding scene are shown in the bottom polar plots. The averaged solar zenith angles for the three scenes are $47.0°$, $45.6°$ and $52.9°$, respectively, as indicated in the polar plots by the red asterisks.





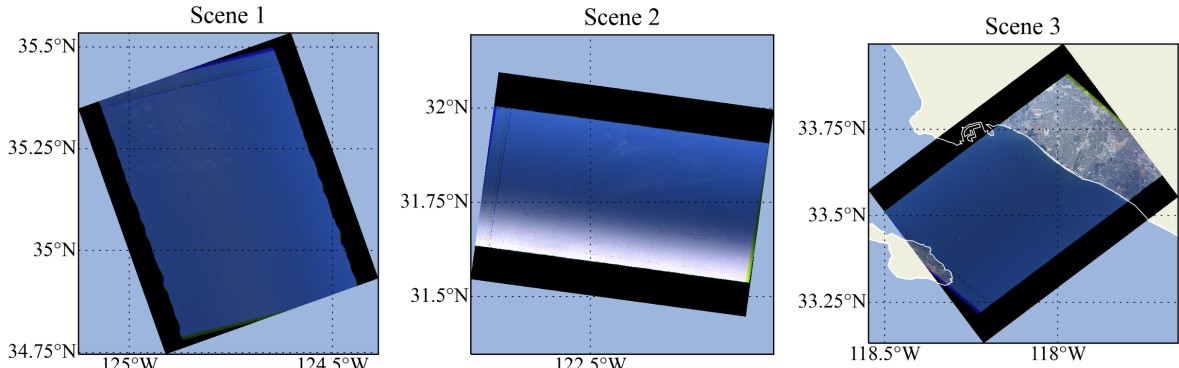

**Figure 13.** The RGB images (670,550,440 nm bands) for the three scenes at near nadir viewing directions. Scene 1 and 2 observe only ocean, while scene 3 observes both ocean and land. Sparse thin clouds are visible from scene 1 and 2.

The HSRL-2 instrument from ACEPOL provided useful aerosol optical depth ground truth at 355 and 532 nm (Hair et al., 2008; Burton et al., 2016), which was used for the validation of the aerosol retrieval algorithm using the AirHARP data. The HSRL-2 measures the pixels along the track as shown in Fig. 12, where an assumed lidar ratio of 40 sr is multiplied by the aerosol backscatter coefficient derived from the HSRL technique to produce aerosol extinction and AOD at 532 nm. For the low aerosol loading cases considered in this study, the assumed lidar ratio approach produces a systematic uncertainty of $\pm 50\%$ (Fu et al., 2020). In scene 3, the aircraft also flew over an AERONET USC_SEAPRISM site, which is equipped with a CIMEL-based system called the Sea-Viewing Wide Field-of-View Sensor (Sea-WiFS) Photometer Revision for Incident Surface Measurements (SeaPRISM) that collects radiances at eight wavelengths of 412, 443, 490, 532, 550, 667, 870, and 1020 nm (Zibordi et al., 2009). AOD from the AERONET data product version 3 level 2.0 data was used in this study, which is also consistent with the HSRL-2 AOD at 532nm as shown latter in Fig. 20. The estimated AERONET AOD uncertainty is from 0.01 to 0.02 with the maximum uncertainty in the UV channels (Giles et al., 2019). We compared AOD from AirHARP retrievals with those from both HSRL and AERONET. Furthermore, to validate the atmospheric correction procedure in the retrieval algorithm, we compared the retrieved remote sensing reflectance with the AERONET ocean color (OC) products as reported by the SeaPRISM measurements at the USC_SEAPRISM site.

Here we applied the FastMAPOL algorithm to the AirHARP field measurements from ACEPOL. The solar and viewing geometries as shown in Fig. 12, and the ozone column density from MERRA2 are the inputs to the forward model. As discussed in Section 2, the total uncertainties are modeled as $\sigma^2 = \sigma_{cal}^2 + \sigma_{avg}^2 + \sigma_{RT}^2 + \sigma_{NN}^2$ for reflectance and DoLP respectively, with all values listed in Table 4.

The histograms of $\chi^2$ for all pixels retrieved in each scene are shown in Fig. 14. Comparing with the histogram in synthetic data retrievals in Fig. 6, the histograms in Fig. 14 show a longer tail with larger $\chi^2$ value. This may due to that some pixels are not represented well by the current forward model, and measurement uncertainties are not well quantified in the cost function as discussed in Sect. 2





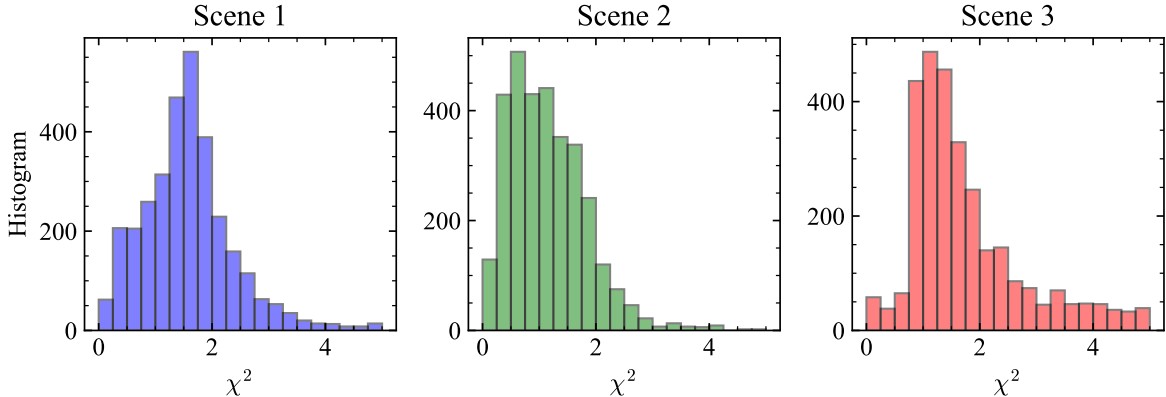

**Figure 14.** The histogram of the cost function values for the pixels in the three scenes as shown in Fig. 12. Each scene has 3283 pixels used in the retrievals. The most probable $\chi^2$ for the three scenes are 1.5, 0.5, and 1.0. A threshold of $\chi^2 < 1.5$ is used in the discussion.

To evaluate the retrieval performance, we plotted the map of the total number of viewing angles used in the retrieval ($N_v$), cost function $\chi^2$, and retrieved AOD(550nm) for each scene in Figs. 15, 16 and 17. As discussed in Sect. 2, the maximum number of viewing angles is 120 for AirHARP measurements. Figs 15,16 and 17 show the number of available viewing angles vary from 0 to 120 due to the removal of glint and other data quality control measures. Discontinuity in the number of angles can be seen as a stripe, due to the removal of angles influenced by water condensation in the lens, which can also be observed in the polar plots in Fig. 12 with the nadir region removed. All three figures show that the number of viewing angles are smaller at the edges parallel to the flight track, where small $\chi^2$ can be achieved but may be less reliable.

For pixels with large $\chi^2$, the forward model cannot fit reflectance or DoLP to the measurements, which may be due to the contamination by cloud (Stap et al., 2015), land, or residuals of glint. The large $\chi^2$ values are also correlated with the large retrieved AOD(550nm) values, for instance, the central region in scene 1. We have excluded retrievals with less confidence and only discussed the retrievals simultaneously satisfying the two criteria with $N_v > 10$ and $\chi^2 < 1.5$. Overall about 40% to 60% pixels are available after applying these two criteria.





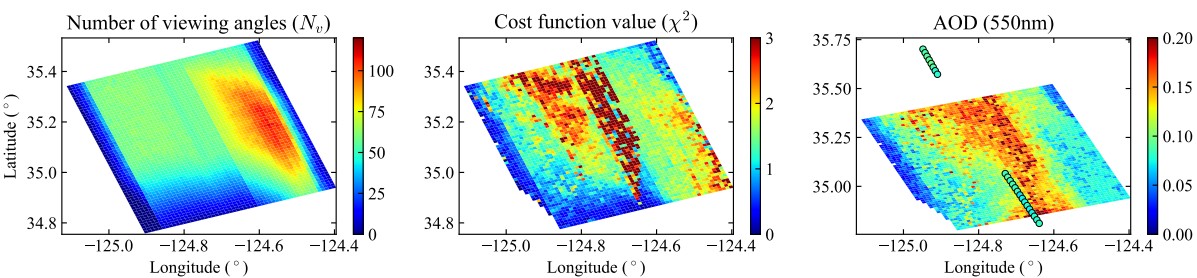

**Figure 15.** The number of viewing angles used in the retrieval($N_v$), cost function value ($\chi^2$) and the retrieved AOD (550 nm) for all pixels in scene 1. Under the condition of $N_v > 10$, 94% pixels are available; with $\chi^2 < 1.5$, 47% pixels are available; 41% pixels under both conditions. The HSRL AOD at 532 nm are indicated by the colored dots in the AOD plot.

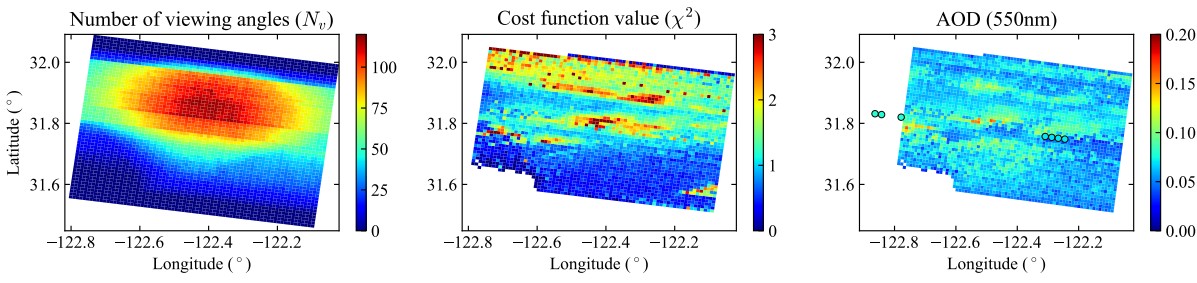

**Figure 16.** Same as 15 but for scene 2, 88% pixels with $N_v > 10$, 67% pixels with $\chi^2 > 1.5$, and 60% pixels under both conditions.

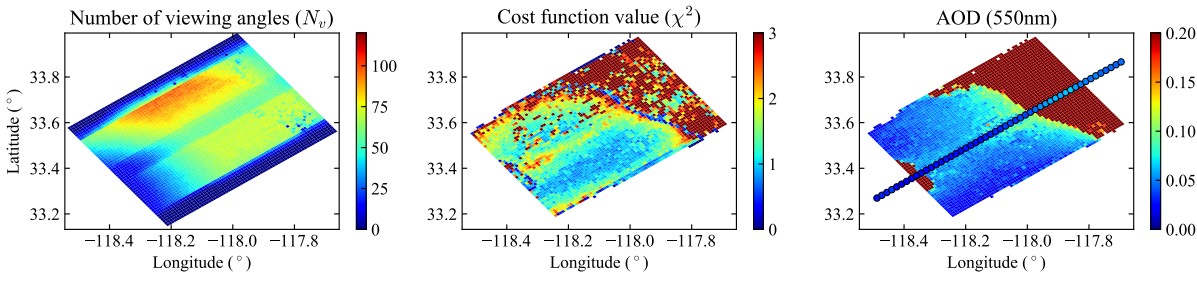

**Figure 17.** Same as 15 but for scene 3, 94% pixels with $N_v > 10$, 47% pixels with $\chi^2 > 1.5$, and 43% pixels under both conditions. The pixels with large $\chi^2$ are mostly influenced by the land (upper region) and island (lower left).

The AOD at 532nm from HSRL data product in all three scenes is in the range from 0.03 to 0.1 as shown in Fig. 18. After applying the two criteria in $N_v$ and $\chi^2$, many pixels from AirHARP retrievals along the HSRL track are not available. To compare with the HSRL AOD along the track direction, the retrieved AOD is averaged in the cross-track direction, the averaged values and its standard deviations are shown in Fig. 18. The averaged and standard deviation of all pixels satisfying the criteria are also shown in the plots. For scenes 1-3, the averaged AOD(550nm) are 0.076, 0.066 and 0.052. The averaged





HSRL AOD is 0.079, 0.072 and 0.038. The average AOD values retrieved from FastMAPOL and those from HSRL agree well for both scene 1 and scene 2. For scene 3, the average AOD value of the FastMAPOL algorithm is larger than the averaged HSRL AOD by 0.014. This may be due to complex water properties not well represented by the open water bio-optical model used in the simulation (Gao et al., 2019). Furthermore, the pixels near the coast are potentially impacted by the adjacency effect

5 of land pixels. However it is challenge to investigate due to the small aerosol loadings. Furthermore, thin cirrus clouds were observed on Oct 23 (Knobelspiesse et al., 2020), which might further impact the accuracy of aerosol retrievals and require future investigations.

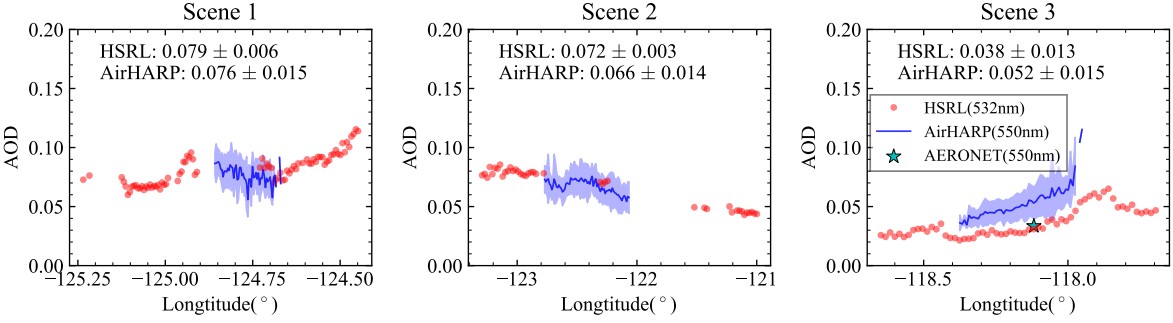

**Figure 18.** Comparison of the retrieved AOD (550 nm) from AirHARP measurement with the AOD (532 nm) from HSRL. The AOD (550 nm) from AERONET USC_SeaPRISM site is shown in scene 3. The AirHARP retrieved AOD is averaged over cross-track pixels with standard deviation as plotted by the shaded areas. The averaged and standard deviations of all pixels for both AirHARP and HSRL are also shown in the texts. Only pixels with $N_v > 10$ and $\chi^2 < 1.5$ are considered.

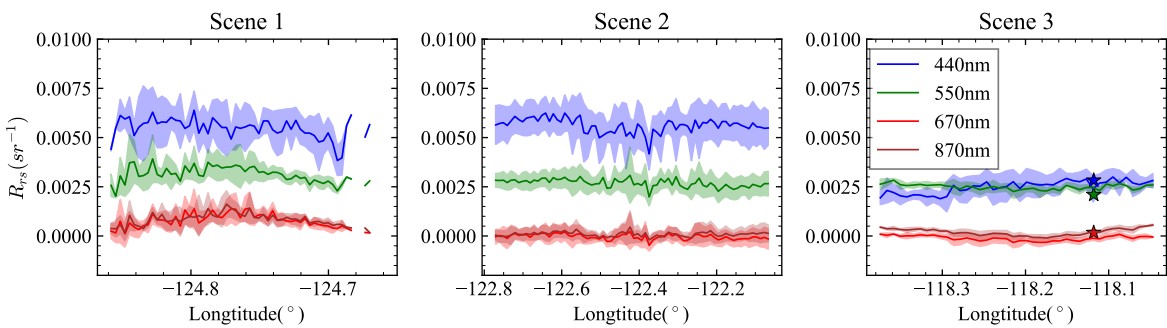

**Figure 19.** The retrieved $R_{rs}$ at the four AirHARP bands. The $R_{rs}$ from AERONET USC_SeaPRISM site are indicated by the star symbols.

Fig. 19 shows the mean value and standard deviation of $R_{rs}$ averaged for the cross-track pixels. $R_{rs}$ from AERONET USC_SeaPRISM site are compared with the retrievals using the nearest wavelengths to the AirHARP bands. The $R_{rs}$ values



for scene 1 and 2 show a larger value of 0.005 to 0.007 at 440nm but reduced to 0.0025 for scene 3, which is closer to the coast. The decrease of $R_{rs}$ may be due to the increase of CDOM as its spectra demonstrated in Fig. 5.

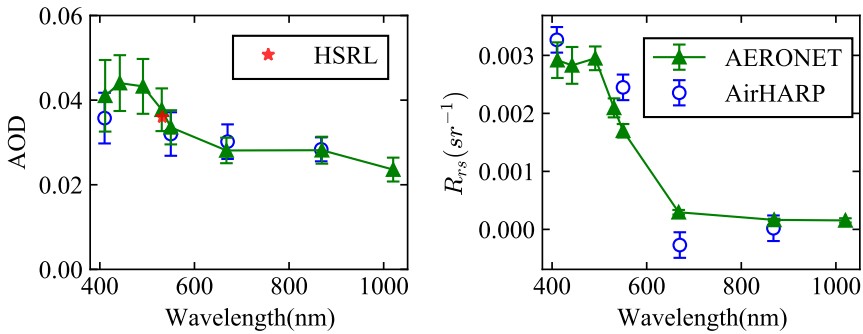

**Figure 20.** The AOD and $R_{rs}$ spectra from AERONET USC_SeaPRISM site on Oct 23, 2017, are shown in green triangles. The error bars indicate the daily average of the AERONET data product. The averaged AOD and $R_{rs}$ from AirHARP retrievals within 1.1km from AERONET site are shown by blue circles.

The AERONET AOD is compared with the FastMAPOL retrieved AOD at 550nm in Scene 3 of Fig. 18, and the whole AOD spectra are compared in Fig. 20. The uncertainties of the AOD and $R_{rs}$ from retrieval are estimated by the average of

2×2 pixels (in 1.1km×1.1 km box) around the AERONET site. The retrieved AODs at 440nm and 870nm agree well with the AERONET results. The difference between the HSRL AOD at 532nm and the AirHARP retrieved AOD near the AERONET site and is smaller than the one shown in the Scene 3 of Fig. 18, which suggests larger differences are contributed by the pixels further away. The retrieved $R_{rs}$ agrees well with the AERONET $R_{rs}$ with a value slightly larger than the AERONET results at 410nm and 550nm by 0.0005 and 0.0006 $sr^1$. The retrieved $R_{rs}$ has a small negative value at 670nm, lower than AERONET

by 0.0007 $sr^1$. Their difference at 868 is 0.0002 $sr^1$. There are larger uncertainties when larger spatial box is considered. Note that this study is done with the AirHARP measurement with 3% uncertainties in the reflectance measurement, which may impact the accuracy in the comparison between the retrieval and in-situ measurements.

## 6    Discussion

The NN model greatly improved the speed of the forward model used in the iterative optimization approach and boosted the

efficiency of the FastMAPOL retrievals. The average retrieval speed for one pixel with FastMAPOL is approximately 2.6 seconds with a single CPU core, or approximately 0.3 seconds with a GPU using the same hardware as mentioned in Sect 3.3. Meanwhile, the NN model maintains a high accuracy as shown in Table 3 and Table 4. For retrieval algorithms running radiative transfer simulation, the accuracy is often tuned down to reduce simulation time. By training a NN, however, the high accuracy of the radiative transfer model simulation can be achieved, as has been demonstrated in this work. Thus, despite the





one time high computational costs in generating the training datasets and conducting the training, the trained NN can be applied efficiently to large observational datasets in the retrieval algorithm.

In the retrieval of the AirHARP field measurement, each pixel has multiple viewing angles that are aggregated from measurements at different times with slightly different solar zenith angles. The NN used in FastMAPOL computes the polarimetric

measurement for specific solar zenith angles for each viewing direction, and therefore, the variation of the solar zenith angle can be captured. These effects may be small for AirHARP measurement in ACEPOL, with a maximum solar zenith angle difference within 0.6 degrees. However, this capability can help to minimize the impacts of the solar angle for HARP2 in space-borne measurements, which can reach to a maximum difference of 1.5° for HARP2 observations.

With the efficient retrievals from FastMAPOL, we have discussed the retrieval performance and uncertainties for the aerosol

properties, including AOD, SSA, refractive index, and particle sizes. Since the AirHARP measurements share many similar characteristics with HARP2 as planned for the PACE mission, the knowledge from the retrieval analysis can help to understand the retrieval performance for the HARP2 instrument in space-borne measurements. Note that HARP2 is likely to have high accuracy due to the onboard calibration capability and the potential to conduct cross-calibration with the OCI instrument. For the development of the NN forward model for space-borne measurements, similar training procedures can be applied

with the sensor altitude at the top of the atmosphere instead of the aircraft altitude used in this study. Due to the impact of retrieval capability by geometry (Fougnie et al., 2020), solar and viewing geometries according to the PACE orbits need to be considered.

The water leaving reflectance is obtained from the atmospheric correction process using the aerosol and ocean properties retrieved from the AirHARP measurements, and a similar procedure can be applied to HARP2 retrievals. Since the hyperspec-

tral OCI in PACE will provide high accuracy measurements, the retrieved information can be applied to OCI and therefore assist hyperspectral atmosphere corrections as demonstrated by Gao et al. (2020); Hannadige et al. (2020). However, aerosol retrieval and atmospheric correction are challenging in the UV spectral range (Remer et al., 2019a). For the ocean bio-optical model in this study, the water properties are modeled as open ocean waters parameterized by a single Chla value. For complex coastal water, complex bio-optical models are preferred in the retrieval of both accurate aerosol properties and water leaving

signals as demonstrated by Gao et al. (2019).

## 7    Conclusions

We have demonstrated the application of a NN for highly accurate forward model calculations of polarimetric measurements for AirHARP. Additional NN models were used to conduct atmospheric correction. These models are used in the FastMAPOL joint retrieval algorithm to conduct simultaneous aerosol property and water leaving signal retrieval. Applications to both the

synthetic AirHARP data and field measurements from ACEPOL are discussed. The uncertainties of the retrieved aerosol properties and remote sensing reflectance are discussed for different aerosol loadings. These results from AirHARP retrievals can help to evaluate the retrieval capabilities for the HARP2 instrument on PACE. In application to field measurements from ACE-POL, the impacts of the number of viewing angles and the value of cost function to the retrieval quality are discussed. Further



comparison with the HSRL and AERONET OC data shows good performance in the retrieval of AOD and remote sensing re-
flectance. Furthermore, the NN forward model and the associated retrieval algorithm enable fast and practical retrievals of the
polarimetric measurement, thus making the algorithm practical for analysis of large data volumes expected from space-borne
imagers such as HARP2. The experience and methodology can be used to help the algorithm development of other satellite
instruments in polarimetric remote sensing.

*Acknowledgements.*  The authors would like to thank the ACEPOL team for conducting the field campaign and providing the data, the Oregon
State University and the University of Southern California teams for maintaining the AERONET USC_SEAPRISM site. M. Gao would like
to thank Frederick Patt, Andy Sayer and George Kattawar for constructive discussions, NASA Ocean Biology Processing Group (OBPG)
system team for support in High Performance Computing (HPC), and NASA Advanced Software Technologies Group (ATSG) for training
in machine learning.

Financial Support:

Meng Gao, Bryan A. Franz, Kirk Knobelspiesse, Brian Cairns, Amir Ibrahim, Joel Gales and P. Jeremy Werdell have been supported by
the NASA PACE project. Peng-Wang Zhai has been supported by NASA (grants 80NSSC18K0345 and 80NSSC20M0227). Funding for the
ACEPOL field campaign came from NASA (ACE and CALIPSO missions) and SRON. Part of this work has been funded by the NWO/NSO
project ACEPOL (project no. ALW-GO/16-09). The ACEPOL campaign has been supported by the Radiation Sciences Program.

*Data availability.*  The data files for AirHARP and HSRL-2 used in this study are available from the ACEPOL website (https://www-
air.larc.nasa.gov/cgi-bin/ ArcView/acepol, ACEPOL team, 2020).

*Author contributions.*  MG, BAF, KK, and PWZ formulated the original concept for this study. MG developed the NN model and generated
the scientific data. PWZ developed the radiative transfer code. KK, PWZ, BC, OH and YH advised on the uncertainty model and retrieval
algorithm. BAF, AI and PJW advised on the atmospheric correction. KK, AI and YH advised on the NN model. JG supported the parallel
computing of the training data. VM, XX, BM and AP provided and advised on the AirHARP data. RF and SB provided and advised on the
HSRL-2 data. All authors participated the reviewing and editing of this paper.

## Appendix A:  Neural networks for AOD and SSA

As summarized in Table 3, we have discussed the NNs used to represent the total reflectance ($\rho_t^f$) and DoLP ($P_t^f$) which are
then used as the forward model in the retrieval algorithm. Using the retrieved aerosol parameters, NNs for $\rho_{t,atms+sfc}^f$ and
$\left[\frac{C_{BRDF}}{T_d t_u}\right]$ are used to compute remote sensing reflectance. To expedite and simplify the calculation of aerosol single scattering





properties such as AOD and SSA as discussed in Section 4, we developed additional four NNs to represent the AOD and SSA for both fine and coarse modes, respectively. These NNs are only used to analyze the retrieved aerosol properties and are not used in the retrieval process. The NN architectures and accuracy are shown in Table A1. The input parameters for the fine mode SSA and AOD are the three submode volume densities, and the real and imaginary parts of refractive index, with a total of 5

parameter. For coarse mode aerosols, there are a total of 4 parameters with only two submodes used. The outputs are the AOD and SSA at the four AirHARP bands.

A total of 10,000 training data points are generated in the same way as in Section 3.1 using the Lorenz-Mie code discussed in Section 2. The NN model accuracy is evaluated with additional 1000 data points not used in the training. As shown in Table A1, the accuracy is much smaller than the retrieval uncertainties shown in Figure 9, therefore the NNs for AOD and SSA

provide sufficient accuracy to evaluate the aerosol single scattering properties.

With the fine and coarse mode AOD and SSA evaluated, the total AOD and SSA can be derived. The total AOD ($\tau_t$) is the summation of the fine and coarse mode AODs as

$$\tau_t = \tau_f + \tau_c \tag{A1}$$

where $\tau_f$ and $\tau_c$ are the fine and coarse mode AODs. The total (or averaged) SSA ($\omega_t$) is defined as the ratio of the total

scattering cross-section and the total extinction cross-section for both fine and coarse modes, which can be computed as

$$\omega_t = \frac{\tau_f \omega_f + \tau_c \omega_c}{\tau_f + \tau_c}, \tag{A2}$$

where $\omega_f$ and $\omega_c$ are the fine and coarse mode SSA.

**Table A1.** The accuracy of the NN for the corresponding quantities in terms of the RMSE ($\sigma$) between the NN predicted values and the truth values from the Lorenz-Mie calculations.

| Quantities | NN architecture | $\sigma$(440 nm) | $\sigma$(550 nm) | $\sigma$(660 nm) | $\sigma$(870 nm) |
|---|---|---|---|---|---|
| AOD(fine) | $5 \times 64 \times 64 \times 4$ | 0.004 | 0.003 | 0.002 | 0.001 |
| AOD(coarse) | $4 \times 64 \times 64 \times 4$ | 0.001 | 0.001 | 0.001 | 0.001 |
| SSA(fine) | $5 \times 64 \times 64 \times 4$ | 0.002 | 0.003 | 0.004 | 0.006 |
| SSA(coarse) | $4 \times 64 \times 64 \times 4$ | 0.01 | 0.01 | 0.01 | 0.01 |



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
