# Peer review of "Efficient multi-angle polarimetric inversion of aerosols and ocean color powered by a deep neural network forward model"

_Atmospheric Measurement Techniques, 2020_

## Referee Comment (RC1)

"Efficient multi-angle polarimetric inversion of aerosols and ocean color powered by a deep neural network forward model" uses a deep neural network (NN) to replace traditional forward model with radiative transfer code as part of the aerosol and ocean color algorithm for multi-angle polarimetric sensor HARP2 and future sensors such as PACE. As more and more machine learning techniques have been implemented in Earth science field, this research is inspiring and informative to the community. The research is well conducted, and the article is nicely organized and written. I have some comments regarding clarification of the procedures and readability of the article.

1. Page 2Line 10 delete "the" before "top-of-atmosphere"
2. Page 4 Line 32 delete "both"
3. Page 5 Line 3 here says "20 viewing angles" but page 3 line 1 says "10 viewing angles"
4. Page 6 Line 1 is there any reference for this "0.01"?
5. Page 7 Line 16, why is imaginary reflective assumed flat among wavelengths? There are many studies show changing absorption as function of AOD, especially over smoke plume. What is the uncertainty related to this assumption?
6. Table 1 It is very confusing, as all parameters are calculated from the forward model, but the description says reflectance measured at BOA/TOO. Are they calculated or measured? Also are these reflectances upwards or downwards?
7. Page 9 line 20. I am not in the field of ocean color, thus, it is confusing for me to read "represented by the spectral remote sensing reflectance" and given Rrs equals to radiance/irradiance. Because reflectance has no unit and Rrs has a unit of sr⁻¹, which I learnt later in the paper in Figure 5.
8. Page 10 Line 3, is the transmittance $t_u^{f,+}$ the same between $\rho_t^f$ and $\rho_t^{f,+}$ vs. $\rho_{t,a+s}^f$ and $\rho_{t,a+s}^{f,+}$? Is $t_u^{f,+} = \frac{\rho_t^f}{\rho_t^{f,+}} = \frac{\rho_{t,a+s}^f}{\rho_{t,a+s}^{f,+}} = \frac{\rho_t^f - \rho_{t,a+s}^f}{\rho_t^{f,+} - \rho_{t,a+s}^{f,+}}$ correct? I assume reflectance at BOA is upward, due to $t_u^{f,+}$ is upward.
9. Page 11 paragraph one, is the height of the aerosol considered? How will NN response when AOD is greater than 0.5, which is often during fire/dust events?
10. Page 12 Line 1 how are these 1000 testing data selected?
11. Figure 4 The y axis labeling is confusing, if the lines are shifted the y axis tick values should be shifted as well to give a right number. Use minor ticks to show the magnitude.
12. Page 16 paragraph one, the percentage error is clearly as a function of reflectance. It can be larger than 3% when reflectance at near-IR is low. Giving one value of percentage error might be misleading, especially when compared with measurement error.
13. Page 17 line 14. "the water leaving signals are represented by the remote sensing reflectance". Again, this sentence is misleading.
14. Page 18 Line 8. The global mean AOD is around 0.2, which means error of AOD is around 14% and it will be even larger when applied to real data due to aerosol assumptions and surface models.
15. Figure 7 and 8, Please add percentage error plots similar to relative error plots shown in the lower panel.
16. Page 22 Line 11, "value magnitude", delete "value" or "magnitude".

17. Page 22 Line 20-24. After I read the next paragraph, I understand better that $R_{rs}$ is the main goal of this procedure, even though AOD and Chla are usually final products.
18. Section 5, HSRL and AERONET data need to be introduced in the beginning of this session especially regarding their accuracies.
19. Figure 14-16, What are the dots on AOD panels? Also please include RGB images for visualization purpose.

---

## Author Comment (AC1)

Response to the report of reviewer 2:

Thank the reviewer for the valuable suggestions and comments, which are very helpful to improve the readability of this work. The comments have been addressed below and the manuscript is revised accordingly.

"Efficient multi-angle polarimetric inversion of aerosols and ocean color powered by a deep neural network forward model" uses a deep neural network (NN) to replace traditional forward model with radiative transfer code as part of the aerosol and ocean color algorithm for multiangle polarimetric sensor HARP2 and future sensors such as PACE. As more and more machine learning techniques have been implemented in Earth science field, this research is inspiring and informative to the community. The research is well conducted, and the article is nicely organized and written. I have some comments regarding clarification of the procedures and readability of the article.

- 1. Page 2Line 10 delete "the" before "top-of-atmosphere" Corrected.
- 2. Page 4 Line 32 delete "both" Corrected.
- 3. Page 5 Line 3 here says "20 viewing angles" but page 3 line 1 says "10 viewing angles" Page 5 Line 3 refers to AirHARP instrument, while page 3 line 1 refers to HARP2 instrument. To make their difference clear, we add a sentence in the manuscript as highlighted below:

"AirHARP measures the total and linearly polarized radiance at 60 viewing angles at the 660 nm band, and at 20 viewing angles at the 440, 550, and 870 nm bands. Different from AirHARP, HARP2 reduces the number of viewing angles to 10 at 440, 550, and 870 nm, and maintains 20 viewing angles at 660 nm, in order to fulfill the bandwidth requirement and preserve information content as much as possible."

4. Page 6 Line 1 is there any reference for this "0.01"? The value is from communication with the HARP team, and is a conservative estimate as noted in the manuscript. A reference is added for the lab calibration:

"AirHARP was calibrated in the lab with an accuracy of 3-5 % for reflectance, and 0.005 for DoLP (McBride et al, IAC 2019). In-flight uncertainty for the AirHARP DoLP is conservatively estimated to be at most 0.01 without an onboard calibrator."

5. Page 7 Line 16, why is imaginary reflective assumed flat among wavelengths? There are many studies show changing absorption as function of AOD, especially over smoke plume. What is the uncertainty related to this assumption?

Thank you for the comments. Aerosols over ocean with spectrally flat spectra, such as sea salt, are common, and can be sufficiently described by our model.

For smoke plume over ocean, having a spectrally constant imaginary index for the visible spectral region assumes that the absorption is caused by black carbon. The primary issue with making this assumption is that it neglects brown carbon. Taylor et al (https://doi.org/10.5194/acp-2020-333) estimate that for their observations of highly aged biomass burning smoke over the SE Atlantic brown carbon contributes ~10% of the absorption at 405 nm. The results of Pistone et al. (https://doi.org/10.5194/acp-19-9181-2019, 2019) are also consistent with absorption being dominated by black carbon with a somewhat smaller contribution from brown carbon. However brown carbon contributions to absorption decrease rapidly with increasing wavelength so we regard the assumption of a spectrally constant imaginary index as reasonable for the measurements analyzed in this study.

6. Table 1 It is very confusing, as all parameters are calculated from the forward model, but the description says reflectance measured at BOA/TOO. Are they calculated or measured? Also are these reflectances upwards or downwards?

Thank you for the comments. All quantities defined in Table 1 are from forward model calculation. We removed the word "measured" in the table to avoid confusion.

The reflectance is defined in Eq (1), which is the ratio of the upwelling radiance and the downwelling solar irradiance.

7. Page 9 line 20. I am not in the field of ocean color, thus, it is confusing for me to read "represented by the spectral remote sensing reflectance" and given Rrs equals to radiance/irradiance. Because reflectance has no unit and Rrs has a unit of sr-1, which I learnt later in the paper in Figure 5.

The reviewer is correct. Rrs has an unit of  $sr^{-1}$ . Since we are discussing both aerosol and ocean color retrievals, we are relying on the terminology established in both fields. We added a few words in the sentence as highlighted below to clarify "remote sensing reflectance" as an defined quantity in ocean color studies:

"An important task for the joint retrievals is to obtain the water leaving signal, which is often represented **in ocean color studies** by the spectral remote sensing reflectance **defined** as  $R_{rs}=L_w^+/E_d^+$ (**Mobley et al 2016**)..."

8. Page 10 Line 3, is the transmittance  $t_u^{f,+}$  the same between  $\rho_t^f$  and  $\rho_t^{f,+}$  vs  $\rho_{t,a+s}^f$  and  $\rho_{t,a+s}^{f,+}$ ? Is  $t_u^{f,+} = \frac{\rho_t^f}{\rho_t^{f,+}} = \frac{\rho_t^f - \rho_{t,a+s}^f}{\rho_t^{f,+} - \rho_{t,a+s}^{f,+}}$  correct? I assume reflectance at BOA is upward, due to  $t_u^{f,+}$  is upward.

Thank you for the discussion.  $t_u^{f,+}$  is the transmittance of the water leaving radiance from BOA to the sensor. To estimate it, we need first compute the water leaving signal at BOA, and the water leaving signal which has reached to the sensor. This is why in Eq(13)

we need to remove the contributions from the ocean surface and atmosphere. The equation  $t_u^{f,+} = \frac{\rho_t^f}{\rho_t^{f,+}} = \frac{\rho_{t,a+s}^f}{\rho_{t,a+s}^{f,+}}$  provided above cannot separate the transmittance for water leaving signal.

9. Page 11 paragraph one, is the height of the aerosol considered? How will NN response when AOD is greater than 0.5, which is often during fire/dust events? The aerosol height is assumed as a constant of 2km as discussed in section 2.1 for the forward model.

Since our NN is only trained with training data within AOD < 0.5, the NN predictions will not be reliable outside that range. Our current research interest is to study both aerosol properties and ocean color. For optically thick aerosol events, it will be hard to observe ocean signals.

10. Page 12 Line 1 how are these 1000 testing data selected? The 1000 testing data points are randomly selected and not included in the training and validation process. Only after the training is done, the testing data is used to evaluate the final NN performance. We revised the sentence for more clarity:

"We **randomly** selected 20,000 cases out of the total 21,000 simulated cases for the training and validation processes, and the remaining 1000 **random** cases will be used as test cases to evaluate the NN accuracy.."

11. Figure 4 The y axis labeling is confusing, if the lines are shifted the y axis tick values should be shifted as well to give a right number. Use minor ticks to show the magnitude. Thank you for the suggestion. We have a sentence in the caption to indicate the shifted y axis. "For each plot, the data points for the 550, 660 and 870nm bands are shifted upward by constant offsets consecutively as indicated by the solid cyan lines"

We attempted to label each shifted y axis with a shifted value, but the y-axis becomes too crowded. One benefit for our current way is that the relative magnitude indicated by the minor ticks still works without explicitly labeling them. We will try to improve if we figure out a better solution.

12. Page 16 paragraph one, the percentage error is clearly as a function of reflectance. It can be larger than 3% when reflectance at near-IR is low. Giving one value of percentage error might be misleading, especially when compared with measurement error. We agree with the reviewer (I cannot locate relevant discussion in Page 16 paragraph one- I believe the reviewer is referring to Page 14 paragraph one). We revised the sentence as follows to indicate the larger percentage uncertainties in 870nm band:

"This translates to a difference for both reflectance and DoLP mostly less than 1%. The maximum percentage difference can be as large as 3% for 870nm bands due to the small reflectance magnitude."

We add further discussions above Table 4 as follows:

"... However, due to the smaller reflectance magnitude at 870nm, the corresponding RMSE for the percentage reflectance difference as shown in Fig.4 is increased from 0.4% at 440nm to 1.0% at 870nm."

- 13. Page 17 line 14. "the water leaving signals are represented by the remote sensing reflectance". Again, this sentence is misleading. We added a reference to the definition of the remote sensing reflectance: "the water leaving signals are represented by the remote sensing reflectance defined in Eq. (10) (Mobley et al 2006)"
- 14. Page 18 Line 8. The global mean AOD is around 0.2, which means error of AOD is around 14% and it will be even larger when applied to real data due to aerosol assumptions and surface models.Thank you for the comments. I cannot locate relevant sentence in Page 18 Line 8, but I believe the reviewer is referring to the results in Figure 9 for the percentage uncertainty of AOD. The reviewer is correct about the uncertainties we obtained in this study.
- 15. Figure 7 and 8, Please add percentage error plots similar to relative error plots shown in the lower panel. Thank you for the suggestions. We feel adding percentage residuals will not add significant new information, but we will attempt to include it in the next revision.
- 16. Page 22 Line 11, "value magnitude", delete "value" or "magnitude". Corrected
- 17. Page 22 Line 20-24. After I read the next paragraph, I understand better that Rrs is the main goal of this procedure, even though AOD and Chla are usually final products. Thank you for the comment. Our approach is to conduct Rrs and aerosol retrieval in a coupled atmosphere and ocean system, and therefore they are interconnected.
- 18. Section 5, HSRL and AERONET data need to be introduced in the beginning of this session especially regarding their accuracies.

The HSRL and AERONET data are introduced in the second paragraph of Section 5. To further address their uncertainties, we added the following sentence for AERONET:

**"The estimated AERONET AOD uncertainty is from 0.01 to 0.02 with the maximum uncertainty in the UV channels (Giles et al. 2019)."**

For HSRL AOD, we added a sentence as highlighted below:

"The HSRL-2 instrument from ACEPOL provided useful aerosol optical depth ground truth at 355 and 532 nm (Hair et al.,2008; Burton et al., 2016), which was used for the validation of the aerosol retrieval algorithm using the AirHARP data. TheHSRL-2 measures the pixels along the track as shown in Fig. 12, where an assumed lidar ratio of 40 sr is multiplied by the aerosol backscatter coefficient derived from the HSRL technique to produce aerosol extinction and AOD at 532 nm. For the low aerosol cases considered in this study, the assumed lidar ratio approach produces a systematic uncertainty  $\pm$ 50 % (Fu et al 2020)."

We further compared HSRL AOD with AERONET AOD to confirm they are consistent in this study:

"AOD from the AERONET data product version 3 level 2.0 data was used in this study, which is also consistent with the HSRL-2 AOD at 532nm as shown latter in Fig. 19."

19. Figure 14-16, What are the dots on AOD panels? Also please include RGB images for visualization purpose.

The dots are the HSRL AOD(532nm). We added the following sentence to the figure caption:

```
"The HSRL AOD at 532 nm are indicated by the colored dots in the AOD plots."
```

Figure 13. The RGB images (670,550,440 nm bands) for the three scenes at near nadir viewing directions. Scene 1 and 2 observe only ocean, while scene 3 observes both ocean and land. Sparse thin clouds are visible from scene 1 and 2.

**Efficient multi-angle polarimetric inversion of aerosols and ocean color powered by a deep neural network forward model**

Meng Gao1,2, Bryan A. Franz1, Kirk Knobelspiesse1, Peng-Wang Zhai3, Vanderlei Martins3, Sharon Burton4, Brian Cairns5, Richard Ferrare4, Joel Gales1,6, Otto Hasekamp7, Yongxiang Hu4, Amir Ibrahim1,2, Brent McBride3, Anin Puthukkudy3, P. Jeremy Werdell1, and Xiaoguang Xu3 1NASA 
[revised manuscript text omitted]

---

## Author Comment (AC2)

Response to the report of reviewer 1:

Thanks to the reviewer for his time and efforts in reviewing this work. The comments and questions are valuable in improving the clarity of this work. We have responses below with manuscript revised accordingly. Improved retrieval results with doubled spatial resolution are updated in Sec 5. We also made the datasets including all AirHARP retrieval results publicly available with a link specified in the manuscript (https://data.nasa.gov/Earth-Science/FastMAPOL_ACEPOL_AIRHARP_L2/8b9y-7rgh).

The authors use a preexisting radiative transfer (RT) code to train a neural network (NN) to output polarimetric radiances at the angles and wavelengths of the AirHARP instrument. The computationally inexpensive NN forward model is then paired with the inversion module of the preexisting MAPOL algorithm to produce a new hybrid retrieval (FastMAPOL) that is less computationally expensive than the original method. The FastMAPOL retrieval is tested on both synthetic data and real ACEPOL measurements made by the AirHARP instrument and, in both cases, good retrieval performance is observed.

Polarimetric remote sensing has the potential to provide enhanced aerosol information but the large number of free parameters in most multiangle polarimeter (MAP) retrieval algorithms generally prohibits the use of precomputed lookup tables. This fact frequently necessitates computationally expensive online radiative transfer calculations which pose a significant challenge for operational algorithms that need to process very large data volumes. The authors provide a convincing demonstration of a machine learning approach that significantly reduces this computational burden. Care is taken to ensure that the NN reproduces the original RT with high fidelity and tests involving synthetic and real data both show good retrieval performance. The approach appears to be technically sound and has the potential to serve as a foundation for retrieval developments pertaining to PACE and other future missions employing MAPs. Furthermore, the content of the manuscript is well organized, and the material is clearly appropriate for AMT. Therefore, once the comments below have been addressed, I can confidently recommend publication.

Thanks for the summary and the positive comments.

**Specific Comments**

1. P2, LN21: The list of MAPS in the sentence beginning on this line seems to have an extra "and". It also is inconsistent in its use of the name of sensor or the overarching mission. I would recommend rephrasing.

   Thank you for the suggestion. We have revised the sentence as follows:

   "Several satellite missions plan to carry MAP instruments, which are scheduled to be launched in the time frame of 2023-2024, including the European Space Agency (ESA)'s Multi-viewing Multi-channel Multi-polarisation Imager (3MI) **on board the Metop-SG satellites** (Fougnie et al, 2018), the National Aeronautics and Space Administration (NASA)'s Multi-Angle Imager for

Aerosols (MAIA) (Diner et al, 2018) and Plankton, Aerosol, Cloud, ocean Ecosystem (PACE) (Werdell et al 2019) missions."

Details on PACE instruments are provided in the next paragraph in the manuscript. .

2. P3, LN22: Did the authors intended to say "...references within)."?

   Corrected.

3. P4, LN2: Citation should be inline (no leading parenthesis)

   Corrected.

4. P4, LN4: Should read "...benefits of using..."

   Corrected.

5. P4, LN32: "have" should be "has"

   Corrected.

6. P5, LN8: It would be clearer to say something along the lines of "...observational altitude..." since not all members of the HARP family are aircraft instruments.

   Thanks. We have revised the phrase as recommended.

7. P6, LN6: AirHARP's swath width aboard the ER2 should be provided.

   Thank you for the suggestion. The following sentence is moved to the first paragraph in Sec 5 with swath width added:

   "AirHARP conducted high spatial resolution measurements with a grid size of 55 m and **swath width of 42km at nadir (up to 60km at far angles).** We averaged the reflectance and DoLP respectively within a bin box of 10 x 10 pixels (550m × 550 m)."

8. P6, LN8: Is σavg the regular standard deviation of all 100 pixels in the box or is it the standard error of the mean (e.g., $\sigma/\sqrt{N}=\sigma/10$)? The latter strikes me as a more appropriate choice as I believe the retrievals are performed on the means of the aggregated 10x10 boxes. The definition of σavg needs to be clarified, and justification for that particular definition should be provided.

   Thank you for the discussion. $\sigma_{avg}$ is used to capture the spatial variation of geophysical properties within the bin box, which provides a good uncertainty indicator for field measurement. This is why we used it in our previous noise model. However, how to include $\sigma_{avg}$ in the noise model depends on the assumption of the noise properties. From

our recent tests, we adjusted our approach and further improved our retrieval performance through 1) quality check of the measurement data,  2) adding a constraint in aerosol model, 3) using $\sigma_{avg}$ to evaluate measurement uncertainty, but not in the noise model. Since $\sigma/sqrt(N)=\sigma/10$ is much smaller than the calibration uncertainty, our new noise model is similar to what the reviewer suggested. A corresponding discussion is provided in Sec 5:

> "To assess the spatial variability of field measurement, we computed the standard deviations for the reflectance ($\sigma_{\rho,avg}$) and DoLP ($\sigma_{P,avg}$) within the bin box. Representative values are provided in Table 5. The values of $\sigma_{\rho,avg}$ and $\sigma_{P,avg}$ at 870nm band are 4.5% and 0.05, respectively, which suggests larger measurement uncertainties at 870nm than other bands probably due to small radiometric magnitudes. Meanwhile, our retrieval tests showed larger coarse mode retrieval uncertainties than synthetic data results. To better constrain retrievals, we assume the coarse mode aerosol as sea salt by setting its imaginary refractive index to zero.  All other retrieval parameter range are kept the same as in Table 2. Furthermore, we found our forward model cannot predict the angular variation of DoLP in 440nm band well (with an estimated MAE of 0.04), which contributes a major portion to the cost function and increases both fine and coarse mode retrieval uncertainties. Therefore, we exclude DoLP in 440nm band from our retrievals in this study."

The corresponding results are updated in Sec 5, and more details are provided in the responses to comments 27-29.

9. P6, LN18: As I understand it, $\sigma NN$ is actually the difference between the RT code and NN, not the NN's uncertainty in an absolute sense. I would recommend clarifying what is meant by "uncertainty" here.

The reviewer is correct. To evaluate the NN accuracy, we generate a synthetic AirHARP datasets with multiple angle setup. This is more realistic than the single angle used in training data. The NN accuracy is then computed by the RMSE between the NN predictions and the synthetic data.  Detailed discussions are in Sec 3.3. We added the following sentence for clarification.

> "$\sigma_{NN}$ is evaluated by comparing with synthetic multi-angle AirHARP measurements discussed in Sec 3.3. "

10. P6, LN32: I generally take "trace gases" to be those gases in the atmosphere other than nitrogen, oxygen, and argon. Was Rayleigh scattering from these three non-trace gases taken into account above the aircraft? Please clarify.

Thank you for the question. Full Rayleigh scatterings is considered. We have revised the sentence to be more precise as follows:

"The atmosphere is configured as three layers: **a top molecular layer above the aircraft,** a molecular layer in the middle below the aircraft, and an aerosol and molecular mixing layer on the bottom with a height of 2 km. Aerosols are **assumed to be uniformly distributed in the mixing layer** as shown in the left panel of Fig. 1. The same vertical structure of the atmosphere was successfully used in the inversion of RSP data (Gao et al 2019, 2020).

**The atmospheric surface pressure is assumed to be one standard atmosphere pressure, which is consistent with the value discussed in Sec 5.  Anisotropic molecular Rayleigh scatterings are accounted (Hansen et al 1974) .**The molecular absorption properties are computed by the hyperspectral line-by-line atmospheric radiative transfer simulator (ARTS) (Buehler et al. 2005) with the molecular absorption parameters of oxygen, water vapor, methane, and carbon dioxide from the HITRAN database (Gordon et al. 2017). The gas absorption of ozone and nitrogen dioxide are from Gorshelev et al. 2014, Serdyuchenko et al. 2014 and Bogumil, K., et al. 2003, respectively. The hyperspectral absorption coefficients are then averaged within the instrument spectral response function and used in the multiple scattering radiative transfer simulation (Zhai et al 2009, 2010, 2018). The molecular profile used is the US standard atmospheric constituent profiles (Anderson et al, 1986)
"

The following references are added:

Buehler, S. A., P. Eriksson, T. Kuhn, A. von Engeln and C. Verdes (2005), ARTS, the Atmospheric Radiative Transfer Simulator, J. Quant. Spectrosc. Radiat. Transfer, *91*(1), 65-93, doi:10.1016/j.jqsrt.2004.05.051.

I.E. Gordon, L.S. Rothman, C. Hill et al., "The HITRAN2016 Molecular Spectroscopic Database", *Journal of Quantitative Spectroscopy and Radiative Transfer* **203**, 3-69 (2017).

Gorshelev, V., Serdyuchenko, A., Weber, M., Chehade, W., and Burrows, J. P.: High spectral resolution ozone absorption cross-sections – Part 1: Measurements, data analysis and comparison with previous measurements around 293 K, Atmos. Meas. Tech., 7, 609–624, https://doi.org/10.5194/amt-7-609-2014, 2014.

Serdyuchenko, A., Gorshelev, V., Weber, M., Chehade, W., and Burrows, J. P.: High spectral resolution ozone absorption cross-sections – Part 2: Temperature dependence, Atmos. Meas. Tech., 7, 625–636, https://doi.org/10.5194/amt-7-625-2014, 2014.

Bogumil, K., et al. (2003), Measurements of molecular absorption spectra with the SCIAMACHY pre-flight model: Instrument characterization and reference

data for atmospheric remote-sensing in the 230–2380 nm region, J. Photochem. Photobiol. A: Chem, 157(2–3), 167–184, doi:10.1016/S1010-6030(3)00062-5.

11. P7, LN18: The term "visible spectrum" should be replaced with something that also includes HARP's 870 nm NIR channel.

Thank you for the suggestion. We have revised the sentence as follows:

"For the application to AirHARP bands, $p_1$ for the real part of the refractive index is approximately spectrally flat for both the fine and coarse mode aerosols **within the AirHARP spectral range**."

12. P7: The vertical profile assumed for the simulated aerosol should be described.

Thank you for the question. The aerosol profile is assumed uniform within 2km range. Please refer to response to comment 10.

13. P9, LN13: Only 14 quantities are listed but it's stated that the forward calculation uses 15 parameters. The 15th parameter should be included in this summary paragraph (presumably ozone column density?).

Thank you for pointing this out. We have added the ozone density in the sentence as follows:

"In summary, the parameters used to represent the forward model include five volume densities (one for each submode), four independent parameters for the refractive indices of fine and coarse modes, one parameter for wind speed, **ozone column density**, and Chla."

14. P9: Should some of the quantities in the equations of section 2.2 have an azimuthal dependence, as well as solar and viewing zenith angle dependence? Why do the equations here show these quantities to be functions of the latter two, but not relative azimuth?

Thank you for the comments. The reviewer is correct. There are dependencies on relative azimuth. We revised the equation by explicitly adding the relative azimuth into the equations (10), (11), (13) and (14).

15. P9: Equation (11) contains an extra parenthesis.

Corrected.

16. P10, LN11: This sentence needs to be reworked so that it is grammatically correct. Also, it should be specified exactly which angle is less than 1° (viewing zenith angle?).

Thank you for the question. Previously sentence is not accurate. We revised the sentence and add more details:

> "To compute the remote sensing reflectance from the multi-angle AirHARP measurement, we only consider the reflectance at the minimum viewing zenith angle for each wavelength and apply the atmospheric correction and BRDF correction as discussed above. For $\theta'_v < 15°$ ($\theta_v < 20°$ ), the $R_o/R$ factor is approximately a constant value of 1, but for larger $\theta'_v$ angles the ratio increases with both wind speed and $\theta'_v$ value (Morel et al 1996, 2002). In this study, we ignored the $R_o/R$ factor in Eq. (14), which will not impact Rrs retrievals from synthetic data due to the small viewing zenith angle used, but may cause underestimation of Rrs at the edge of the image, as will be discussed in Sec 4 and 5."

Sec 4 added more details regarding synthetic data:

> "Note that the Ro/R factor will not impact the BRDF correction in computing Rrs for the synthetic data, because of the small viewing zenith angle used at the four AirHARP bands ,which are $1.22°,1.17°,0.03°,3.52°$ respectively "

Further discussions are added for field data (please also refer to response to comment 20):

> "As discussed in Sec 2.2, we chose the minimum viewing zenith angle available from the measurements after removing the sunglint. The removal of sunglint improves the Rrs calculation for Scene 2 as shown in Fig. 16. Moreover, we ignored the R0/R factor in Eq. 14 which may cause underestimation of the Rrs at the edge of the image where $\theta_v$ can reach as large as 60°. However, it is challenging to analyze its impact at large $\theta_v$ angles. R0/R has a strong dependency on wind speed, but the retrieved wind speeds from current retrievals show large uncertainties. Further work may require a better treatment of sunglint and improved accuracy in wind speed."

17. P11, L2: A more precise description of the method of random sampling should be given. Are the variables being drawn from a uniform distribution (log-uniform distribution in the case of Chla)?

Thank you for the suggestion. We revised the sentence as follows:

> "The solar zenith angle, ozone column density, refractive index, and wind speed are randomly sampled **from a uniform distribution**. Chla is randomly sampled from **a log-uniform distribution**."

18. P12, LN14: Here it is stated that separate NNs are used for reflectance and DoLP. Instead of two separate NNs, another potential approach would have been to use a single NN with an 8-dimensional output (4 reflectance + 4 DoLP values). It would be beneficial if the authors could further elaborate on the pros and cons of these two possible approaches, and their motivation for using the particular 2 network architecture that was ultimately chosen.

Thank you for the questions. We choose the current approach with the consideration of both flexibility and efficiency. Combining two NNs together may lead to better efficiency. However, there are several challenges with this strategy: 1) as shown in the Figure 2 and 3, reflectance and DoLP differ in angular variations which may increase the difficulties in training the neural network without increasing the size of NN: 2) there are different accuracy requirements for reflectance and DoLP, which is easier to control when the two NNs are separated. There could be other ways to combine reflectance with DoLP such as working with I, Q, U directly, which we will explore in future work.

We added the corresponding discussions into Sec. 3.1:

"

To maintain both flexibility and efficiency, we trained two NN models for reflectance and DoLP respectively in the next section. Reflectance and DoLP have different accuracy requirements as discussed in Sec. 2, and also differ in angular variations as shown in the Figs. 2, therefore, it is convenient to control their accuracy through separated training procedures.
"

19. P19, LN 5: Is the added noise uncorrelated in angle, polarization and wavelength? If so, how would the authors expect more realistic calibration errors (which likely will be strongly correlated among these three dimensions) to impact their results?

Thank you for the question. Yes, we assume no correlation in the added noise. We conducted an independent study in which noise was added with correlation in angles, and observed overfitting in retrievals (reduction of $\chi^2$). This is related to the fact that some noise variations are long angular range in nature, which can be similar to some true signals and thus get fitted by the optimization process. The quantified influence depends on the magnitude of the noise and the strength of the correlation (Knobelspiesse OE 2012), which requires further study.

20. P22: There is an extra parenthesis in the title of the Chla subplot of Figure 9.

Removed.

21. P22, LN5: The last two sentences of this paragraph have several grammatical errors and are a bit confusing. I suggest rephrasing.

We revised the sentence as follows:

> "…For wind speed larger than 3 m/s, the averaged retrieval uncertainty is 1.2 m/s with a small variation less than 0.1 m/s. "

22. P23, LN14: Should read "...less sensitive to outliers." (No "the")

Corrected.

23. P23, LN14: "...applied to the synthetic measurements..."

Corrected.

24. P23, LN23: The sentence starting on this line has several grammatical errors and needs rewriting.

We revised the sentence as follows:

> "For 440 nm band, when AOD is less than 0.3, the Rrs retrieval uncertainties are less than 0.0005 sr$^{-1}$; but the uncertainties become as high as 0.001 sr$^{-1}$ at a larger AOD of 0.5".

25. P23, LN28: It would be clearer if the phrase "the reduction of" was changed to "reduce".

Revised.

26. P25, FIG12: The figure would be easier to interpret if the caption explained that the "hole" in the middle of the three polar plots was due to water condensation on the lens. I thought it was a plotting artifact, as it lines up the "0°" label, until I read the full condensation explanation later in the text. It might even be good to remove the "0°" tick completely.

In addition to the explanation in the main text, we added a sentence in the caption:

> " The central portion in the viewing angle plots is removed due to water condensation on the lens."

We prefer to keep the 0 degree symbol to ensure readers understand the central point refers to nadir (with no removal of zenith angles in the plots).

27. P26, LN20: The last sentence needs to have grammatical errors fixed and its meaning clarified.

As discussed for comment 8, we have improved the retrieval performance with a new $\chi^2$ histogram shown below. The discussion of $\chi^2$ is combined with a latter sentence as discussed for the next comment. We revised the sentence as follows:

"The histograms of $\chi^2$ for all pixels retrieved in each scene are shown in Fig. 14. The most probable $\chi^2$ are 1.1, 1.7, and 1.1 respectively."

[Figure]

*Figure 14.The histograms of the cost function values over the three scenes as shown in Fig. 12 with total pixel numbers of 13491, 13226,and 9159. Only pixels over the ocean are considered.*

28. P27, LN5: More discussion of the along-track stripping in $\chi^2$ values should be provided. In many cases, it does not seem to correlate with the number of view angles available. For example, the most prominent ~10 pixel wide dark red strip of $\chi^2$ in Figure 15 is actually offset to the right by about five pixels of the band of missing angles at nadir. What is the cause of this artifact?

Thank you for checking the images and questions. We investigated the stripes, and found anomalies from both measurements and retrieved results, which have been corrected (Fig 15). Meanwhile, we have updated the retrievals as discussed in comment 8. The spatial resolutions are also doubled. The new plots are shown below for Figure 15-17. Note that due to the use of smaller uncertainties in the noise model (without contribution from $\sigma_{avg}$), the overall value of $\chi^2$ increased, but we still have a majority of pixels converging to within $\chi^2 < 2$~3.

More discussion was added to the paragraph (also see response to comment 30):

"For pixels with larger $\chi^2$ as shown in Fig. 14, the forward model cannot fit the measured reflectance or DoLP well, which may be due to cloud contamination (Stap et al 2015), land, or residuals of glint. In scene 1, the top region with large $\chi^2$ values is mostly due to the impact of thin cloud which is visible from Fig. 13. Larger residuals in the 870nm band between measurement and forward model are also observed. The retrieved AODs are over-estimated in this region. In scene 2, the region with $\chi^2$ >3 correlate closely with the thin clouds (Fig 13), which influence nearby AOD and Rrs retrievals. For scene 3, $\chi^2$ becomes larger than 2 when close to the coast. This may be due to complex water properties which are not well represented by the open water bio-optical model used in the simulation

(Gao et al 2019). The pixels near the coast are also potentially impacted by the bottom effect and adjacency effect of land pixels."

[Figure]

Fig. 15.The number of viewing angles used in the retrieval (Nv), cost function value ($\chi^2$), the retrieved AOD (550 nm) and Rrs (550nm) for all pixels in scene 1. The HSRL AODs at 532 nm are indicated by the colored dots in the AOD plot.

[Figure]

Figure 16. Same as Fig. 15 but for scene 2. For Rrs, viewing angles at least 40° away from the solar specular reflection direction are used to avoid sunglint as shown in Fig. 13.

[Figure]

Figure 17. Same as Fig. 15 but for scene 3. The pixels with large $\chi 2$ are influenced by the land (upper region) and island (lower left). The retrieved AOD and Rrs over land pixels are not shown. The location of the AERONET USC_SEAPRISM site is indicated by a red star symbol.

29. P28, FIG15-17: If it is easy to add, it would be informative to include a fourth subplot with the retrieved Rrs in these figures.

Thank you for the suggestion. We have added the Rrs figures as shown above.

30. P29, FIG18-19: I'm struggling to see the value in these two figures given the very coarse across-track averaging. Why resolve along-track data at ~0.5km when averaging across-track at tens of km? In my mind, a better approach might be to average all pixels within a circle of some radius of the HSRL/AERONET/SeaPRISM measurements. Alternatively, these figures could be removed altogether, while still reporting the average values for each of the 3 scenes. (Much of the pixel-level information is already conveyed in figures 15-17).

Thank you for the suggestion. We agree with the reviewer, the original comparison is not ideal, but they are useful to make clear comparison with HSRL. To fix the above-mentioned issue, 1) we conducted new retrievals with a higher resolution, 2) pixels in the along track direction with 4x4 pixels are averaged, and 3) we kept most pixels with a broader criterion in the plots. Both Fig 18 and 19 are updated and shown below. Discussion for AOD are updated in the paragraph:

> "**To compare with the HSRL AOD in the along track direction, the retrieved AOD (550nm) is averaged within a box of 2.2km x 2.2km (4 x 4 pixels). The averaged AOD (550nm) values and the corresponding standard deviations are shown in Fig. 18.** Pixels with Nv>30 and $\chi^2$ <10 are considered. The overall averaged values and their standard deviation are also computed and indicated in the plots. The averaged HSRL AODs are 0.079, 0.071 and 0.037 for scenes 1 to 3. The averaged retrieved AOD(550nm) are 0.096, 0.078 and 0.049 with relatively larger retrieval variation of 0.02 to 0.03. For scene 1, most $\chi^2$ values are larger than 2, while for the other two scenes, most are less than 2 except for those pixels very close to cloud and coast. The retrieved AOD is larger than that of the HSRL by 0.03 in the overlapped region which may be influenced by the remaining effect of water condensation. In scene 2, the peaks of the retrieved AOD values correspond to the $\chi^2$ values larger than 2, which are influenced by the nearby thin cloud. There are no overlapped pixels except the one associated with high AOD peaks, but the general trend of the retrieved AOD agrees with the HSRL results. For scene 3, the retrieved AOD values agree well with the HSRL AOD with an average difference less than 0.01 and $\chi^2$ mostly less than 2. However when the pixels are close to the coast, both $\chi^2$ and AOD increased significantly as discussed previously."

[Figure]

*Fig. 18 Comparison of the retrieved AOD (550 nm) from AirHARP measurement with the AOD (532 nm) from HSRL for Scenes 1 to 3. The AOD (550nm) from AERONET USC_SeaPRISM site is shown in scene 3. The AirHARP retrieved AOD is averaged with 4x4 pixels (2.2km x2.2km). The averaged and standard deviations for both AirHARP retrievals and HSRL products are provided in the text. Pixels with Nv>30 and $\chi 2<10$ are considered.*

Discussions for Rrs in Figure 19 are also updated:

"Fig. 19 shows the mean value and standard deviation of Rrs averaged in the same way as AOD discussed above. There is similar spatial variation between the retrieved Rrs and AOD. Pixels with large Rrs uncertainties are mostly associated with the large AOD uncertainties shown in Fig. 18. The Rrs values at 440nm for the three scenes are 0.0055, 0.0072, and 0.0030sr$^{-1}$, where the decrease of Rrs from scene 2 to scene 3 may be due to the increase of CDOM close to the coast as demonstrated in Fig. 5. Moreover, Rrs at scene 1 are likely to be under-estimated due to the large $\chi^2$ and retrieved AOD over the center of scene 1. The averaged Rrs values at 550nm remain approximately constant with a value of 0.0003 sr$^{-1}$ over all three scenes. Rrs from AERONET USC_SeaPRISM site are indicated in scene 3 of Fig 19 and also compared in Figure 20. …"

[Figure]

*Figure 19. Similar to Fig. 18, the retrieved Rrs are computed for the AirHARP band of 440, 550, and 670nm bands. The averaged Rrs and its standard deviation are shown in the legends. For scene 3, Rrs from AERONET USC_SeaPRISM site at wavelengths corresponding to AirHARP bands are indicated by the star symbols.*

The comparison with the AERONET (Fig 20) is updated correspondingly:

"To better compare with AERONET results, we only considered the pixels with $\chi 2<2$ and conducted the same averaging ( 4 x 4 pixels) around the USC_SeaPRISM site for the retrieved AOD and Rrs. The averaged values and their standard deviations are plotted in Fig. 20. The overall retrieved AOD spectrum is similar to AERONET results with a difference smaller than 0.01. The results are similar to the retrieval results from the Research Scanning Polarimeter as reported by Gao et al. (2020). The retrieved Rrs agrees well with the AERONET Rrs with a difference less than 0.001 sr$^{-1}$. Note that this study is done with possible AirHARP measurement uncertainty of 3% in reflectance, which may impact atmospheric correction accuracy."

[Figure]

*Fig 20: Comparisons of the AOD and Rrs from AirHARP retrievals with AERONET products. The retrieval results are averaged with 4×4 pixels (2.2km×2.2 km) around the AERONET USC_SeaPRISM site. This is similar to Fig. 5 and 19 with error bars indicating the standard deviations, but only pixels with χ2<2 are considered. The AERONET AOD and Rrs spectra are taken from Oct 23, 2017 with the error bars indicating daily variations. HSRL AOD at 532nm is also shown.*

31. P30, LN16: Was the speed up factor of FastMAPOL every specified? How long does a retrieval using equivalently accurate RT take with regular MAPOL on the same hardware?

Thank you for the questions. We added the information:

"Comparing to the retrieval speed of approximately 1 hour per pixel using conventional radiative transfer forward model, the computational acceleration is $10^3$ times faster with CPU or $10^4$ times with GPU processors."

32. P21, LN12: I think the authors intend to say "...HARP2 is likely to have high*er* accuracy..."

Thank you. I believe the author is referring to P31. We have corrected the sentence.

Furthermore, we have provided the complete dataset from AirHARP retrievals with brief discussion in Sec 5:

[revised manuscript text omitted]